# Multiplexed characterization of rationally designed promoter architectures deconstructs combinatorial logic for IPTG-inducible systems

Timothy C. Yu[1,14], Winnie L. Liu[2,14], Marcia S. Brinck [3], Jessica E. Davis[4], Jeremy Shek [4], Grace Bower[2], Tal Einav [5], Kimberly D. Insigne[6], Rob Phillips[5,7,8], Sriram Kosuri [4,9,10,11,12,13,15] & Guillaume Urtecho [13,15]

A crucial step towards engineering biological systems is the ability to precisely tune the genetic response to environmental stimuli. In the case of *Escherichia coli* inducible promoters, our incomplete understanding of the relationship between sequence composition and gene expression hinders our ability to predictably control transcriptional responses. Here, we profile the expression dynamics of 8269 rationally designed, IPTG-inducible promoters that collectively explore the individual and combinatorial effects of RNA polymerase and LacI repressor binding site strengths. We then fit a statistical mechanics model to measured expression that accurately models gene expression and reveals properties of theoretically optimal inducible promoters. Furthermore, we characterize three alternative promoter architectures and show that repositioning binding sites within promoters influences the types of combinatorial effects observed between promoter elements. In total, this approach enables us to deconstruct relationships between inducible promoter elements and discover practical insights for engineering inducible promoters with desirable characteristics.

[1] Department of Bioengineering, University of California, Los Angeles, CA 90095, USA. [2] Department of Molecular, Cell, and Developmental Biology, University of California, Los Angeles, CA 90095, USA. [3] Department of Microbiology, Immunology, and Molecular Genetics, University of California, Los Angeles, CA 90095, USA. [4] Department of Chemistry and Biochemistry, University of California, Los Angeles, CA 90095, USA. [5] Department of Physics, California Institute of Technology, Pasadena, CA 91125, USA. [6] Bioinformatics Interdepartmental Graduate Program, University of California, Los Angeles, CA 90095, USA. [7] Division of Biology and Biological Engineering, California Institute of Technology, Pasadena, CA 91125, USA. [8] Department of Applied Physics, California Institute of Technology, Pasadena, CA 91125, USA. [9] UCLA-DOE Institute for Genomics and Proteomics, Los Angeles, CA 90095, USA. [10] Institute for Quantitative and Computational Biosciences (QCB), University of California, Los Angeles, Los Angeles, CA 90095, USA. [11] Eli and Edythe Broad Center of Regenerative Medicine and Stem Cell Research, University of California, Los Angeles, Los Angeles, CA 90095, USA. [12] Jonsson Comprehensive Cancer Center, University of California, Los Angeles, CA 90095, USA. [13] Molecular Biology Interdepartmental Doctoral Program, University of California, Los Angeles, CA 90095, USA. [14] These authors contributed equally: Timothy C. Yu, Winnie L. Liu. [15] These authors jointly supervised this work: Sriram Kosuri, Guillaume Urtecho. ✉email: sri@ucla.edu; gurtecho@ucla.edu

nducible promoters are key regulators of cellular responses to external stimuli and popular engineering targets for applications in metabolic flux optimization and biosensing[1–3]. For example, inducible systems have been designed to function as controlled cell factories for biosynthesis as well as non-invasive diagnostics for gut inflammation[4,5]. However, these applications generally rely on synthetic inducible promoters that can elicit precisely programmable responses, a quality that is not exhibited by native promoter systems. As a result, there is a demand for strategies to engineer inducible promoters with desirable characteristics, such as minimal expression in the uninduced state (minimal leakiness) and maximal difference between the induced and uninduced states (maximal fold change). More broadly, the design and analysis of synthetic inducible promoter function provide insight on the biophysical processes driving gene regulation.

A variety of approaches have been implemented to engineer inducible promoters, however, these strategies have their shortcomings. Previous studies have had great success implementing biophysical models to tune the relative behaviors of regulatory elements and explain promoter expression, but do not tell us how the repositioning of binding sites influences expression[6–10]. Directed evolution is a promising strategy that leverages stepwise random mutagenesis and selection to identify favorable promoters, but is generally limited to optimizing within local, evolutionarily accessible sequence space[11,12]. While this black box approach can produce variants with the desired phenotype, it often requires iterative rounds of library screenings[12] and does not inform our ability to logically construct promoters. Lastly, rational design is a promising approach based on the application of pre-existing mechanistic knowledge of inducible systems to generate novel variants[13,14]. Although there is great potential in rationally designed promoters for achieving specific applications, this approach requires a fundamental understanding of how to engineer these systems.

Inducible promoters consist of *cis*-regulatory elements that work in concert with multiple *trans*-acting factors to determine overall expression output[15,16]. As such, a critical step towards learning how to engineer these systems is to interrogate the combinatorial regulatory effects between promoter-based elements. Years of studies on the inducible *lacZYA* promoter have revealed many sequence-based factors influencing its regulation. First, the binding affinities of operator sites are critical elements in determining the activity of the repressor protein, LacI[17,18]. Second, the nucleotide spacing between operator sites is vital as looping-mediated repression is dependent on repressor orientation[17,19]. Third, the positioning of the repressor sites relative to the RNA polymerase (RNAP) binding sites determines a variety of repression mechanisms and transcriptional behaviors[13,14]. Fourth, the strength of the core promoter modulates RNAP avidity and thus gene expression[6]. However, while previous studies have characterized these modular sequence components individually, the combinatorial effects of these features on promoter induction have yet to be explored.

Inspired by previous success in studying the combinatorial logic of *E. coli* promoters[20], we sought to address these obstacles by integrating rational design with a high-throughput screening of large DNA-encoded libraries. The recent development of massively parallel reporter assays (MPRAs) provides a framework for leveraging next-generation sequencing to measure cellular transcription levels of large numbers of DNA sequence variants. This approach enables the measurement of thousands of synthetic sequences in a single, multiplexed experiment, often using transcriptional barcodes as a readout[20,21]. Previously, this paradigm has also been used to empirically examine both the individual and combinatorial effects of transcription factor binding sites on gene

expression in eukaryotes, improving our ability to design synthetic eukaryotic promoters with programmable responses[22–29]. However, there have been few similar high-throughput studies in prokaryotes.

In this work, we implement a genomically-encoded MPRA to interrogate thousands of rationally designed variants of the *lacZYA* promoter and investigate the relationships between inducible promoter components across four *cis*-regulatory sequence architectures. We first explore the relationship between operator spacing and repression at the *lacUV5* promoter using a variety of transcriptional repressors. Next, we design and characterize 8269 promoters composed of combinations of LacI repressor and RNAP-binding sites, exploring combinatorial interactions between elements and establishing relationships that guide transcriptional behavior. Lastly, we isolate and further characterize promoters with various levels of fold change and leakiness that may be useful in synthetic applications.

## Results

**Repression by transcription factors is dependent on operator spacing.** The *lacZYA* promoter is a classic model for gene regulation in *E. coli*, with many studies investigating the relationship between sequence composition and induction properties. This promoter contains two LacI dimer sites positioned at the proximal +11 and distal −82 positions relative to the transcription start site (TSS)[30,31], which flank a set of σ70 −10 and −35 elements (Fig. 1a, see *WT PlacZYA*). RNAP cooperatively binds these σ70 hexameric sequences and the relative binding affinity of these elements determines promoter strength[6,8]. Conversely, the LacI operator sites repress expression from the native *lacZYA* promoter when bound[32]. While LacI repressor bound at the proximal site blocks RNAP binding as well as promoter escape, binding at the distal site alone does not inhibit transcription and serves a more nuanced role in repression[33]. When both the proximal and distal sites are bound, LacI dimers at these sites can engage in homotetrameric protein interaction, tethering these sites together and forming a local DNA loop[18,34,35]. This repression loop further occludes RNAP binding, decreasing gene expression.

Studies exploring the formation of this repression loop have found that it is heavily dependent on the spacing between LacI operator sites (Fig. 1b)[36–38]. Due to the helical nature of B-form DNA, which completes a full rotation roughly every 10.5 bp, as operator sites are placed at various distances from one another along with the DNA their relative orientation along the face of the DNA helix changes as well. As a result, the ability of the distal site to engage in this repression loop fluctuates as it is shifted along with the promoter, with repression strength correlated with helical phasing between the two operator sites[36,37]. In our effort to optimize the *lacZYA* promoter, we sought to validate the effect operator spacing has on repression, as well as explore whether other repressors follow this same phenomenon.

Accordingly, we tested the relationship between spacing and repression for six transcription factors (TFs) at the most commonly utilized *lacZYA*-derived promoter, *lacUV5*: LacI, AraC, GalR, GlpR, LldR, and PurR. While LacI[35,37,39], AraC[40,41], and GalR[42–46] have been experimentally shown to engage in DNA looping, there is evidence that GlpR[47], LldR[48], and PurR[34] may also be capable of this mechanism. Using reported, natural binding sites for these TFs[49] (Supplementary Table 1), we designed 624 sequences assessing the ability of these sites to repress a constitutive *lacUV5* promoter across various operator spacings. The *lacUV5* promoter models the *lacZYA* canonical architecture, but has a small 2 bp mutation in the −10 to drive more detectable levels of expression[50]. In our design, a proximal

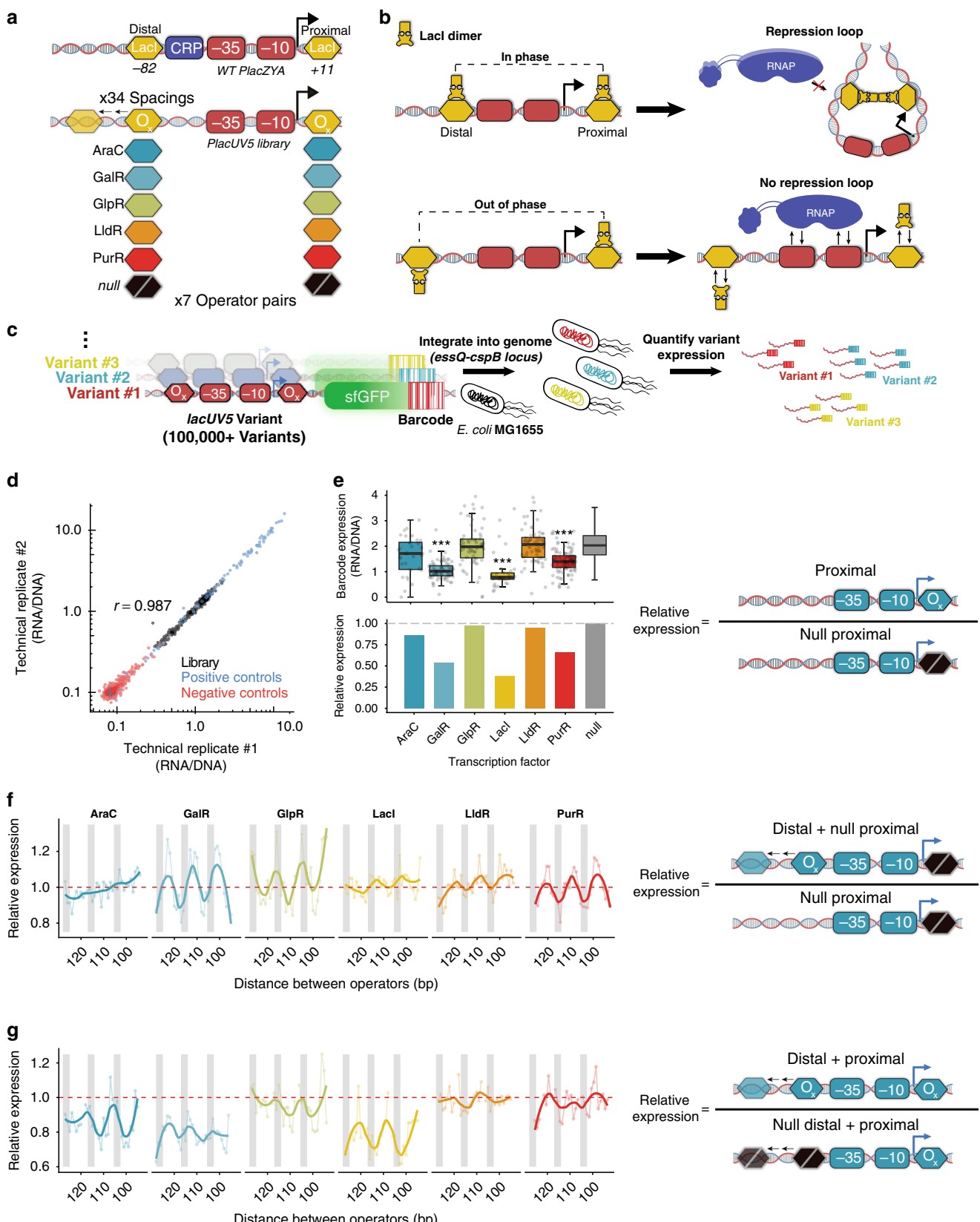

site for each TF was centered at +12, to avoid overlapping the transcription start site, and a series of variants were created in which the distal operator site was centered at each position from −83 to −116 relative to the TSS (Fig. 1a). Furthermore, to quantify the effect of the individual sites, we tested variants where

either the proximal or distal site was replaced with a scrambled sequence variant that maintained the GC content of the native LacI site. We grew this library in MOPS rich-defined media supplemented with 0.2% glucose, a condition for which all TFs should be repressive, and measured expression of all variants

**Fig. 1 Identifying optimal spacing for repressors at *lacUV5* promoter. a** We designed a library of *lacUV5* variants modeled after the WT *lacZYA* promoter. In this library, we evaluate repressor effects when the distal site is moved 32 nucleotides upstream at 1 bp increments. **b** If repressors bind along the same face of the DNA helix, repression loop formation may occur, thereby preventing RNAP association with the promoter. **c** In this MPRA format, pooled promoter variants are engineered to express uniquely barcoded sfGFP transcripts, singly integrated into the *essQ-cspB* locus of the *E. coli* genome. After integration, individual promoter expression was determined en masse using the ratio of the barcode reads from RNA-seq to that of DNA-seq. **d** Comparison of MPRA expression measurements between biological replicates grown in MOPS rich-defined medium supplemented with 0.2% glucose ($r = 0.987$, $P < 2.2 \times 10^{-16}$, two-sided Student's *t* test). **e** MPRA expression when a proximal site is added relative to expression of *lacUV5* without repressor sites. Top shows the distribution of expression for all barcodes associated with each variant, whereas the bottom shows the averaged variant expression relative to *lacUV5* without repressor site (null). Significance levels determined by Welch's two-sided *t* test, ***$P \leq 0.001$. AraC: $n = 35$, $P = 0.07$; GalR: $n = 82$, $P = 6.68 \times 10^{-15}$; LacI: $n = 35$, $P = 2.22 \times 10^{-7}$; LldR: $n = 68$, $P = 0.47$; PurR: $P = 8.973 \times 10^{-7}$. In each boxplot, the lower, middle, and upper hinges correspond to the first quartile, median, and third quartile, respectively. Whiskers represent 1.5× IQR from the lower and upper hinges. **f** Relative MPRA expression as each distal site is moved upstream in the absence of a proximal site relative to *lacUV5* without repressors. Thick lines denote the fit using locally weighted polynomial regression. Thin lines connect data points at sequential intervals. Gray bars indicate 3 bp windows where the distal site is positioned in-phase with the +11 proximal site[17]. **g** MPRA expression as the distal site is moved upstream when the proximal site is present relative to expression of the proximal-only variant. Source data are available in the Source Data file.

using a previously described MPRA[20] (Fig. 1c). In brief, we synthesized each variant and engineered these promoters to express uniquely barcoded GFP transcripts. Using recombination-mediated cassette exchange[51], each barcoded variant was singly integrated into the *essQ-cspB* intergenic locus of the *E. coli* genome, positioned near the chromosomal midreplichore. We then grew the integrated libraries in rich, defined media, and quantified relative barcode expression levels by performing RNA-Seq of the transcribed barcodes and normalizing transcript levels to DNA copy number as determined by DNA-Seq. Using this assay, we recovered expression measurements for 615 (98.6%) of the variants we designed, measuring an average of 70 unique barcodes per variant (Supplementary Fig. 1). These measurements exhibited a high degree of correlation between technical replicates (Fig. 1d, $r = 0.987$, $P < 2.2 \times 10^{-16}$, two-sided Student's *t* test).

We first explored the ability of these TFs to repress the *lacUV5* promoter when placed in the proximal position. To evaluate this, we compared the relative expression between variants with proximal sites to the *lacUV5* promoter containing a scrambled LacI site in the proximal position (Fig. 1e). At this position, repression varied across operators although the AraC, LldR, and GlpR sites were ineffective (AraC: $P = 0.06$, LldR: $P = 0.47$, GlpR: $P = 0.5837$, Welch's two-sided *t* test). LacI exhibited the strongest level of repression in the proximal position at 2.62-fold ($P = 2.22 \times 10^{-7}$, Welch's two-sided *t* test), which may be due to the strong binding affinity of the native proximal operator site[30].

To gauge the performance of these repressors at each position in the distal site, we looked at how expression changes as a function of distance from the proximal site. While LacI[37] and AraC[40,41] are known to exhibit a cyclic pattern of repression as the distance between operator sites is increased, there are no direct measurements showing that GalR, GlpR, LldR, or PurR share this phenomenon. First, we looked at the effect of moving the distal site across 33 nucleotides in the absence of a functional proximal site (Fig. 1f). We observed a uniformity of cyclic behavior across most repressors tested, suggesting position-dependent effects are a general phenomenon of many TFs. Lone GalR, GlpR, and PurR distal sites alternated between activation and repression, a phenomenon which has been observed in similar translocations of a LacI-binding site upstream of a promoter in the absence of inducer[37]. This may be due to helical positioning of the repressor relative to RNAP and resulting steric interference or binding-induced DNA distortions[52]. Interestingly, we observed TFs exhibiting opposing position-dependent behaviors, where GalR and PurR repressed when the center of the binding sites was approximately in-phase with the +1 TSS position and activated when out of phase. We observed the

opposite effect with GlpR sites. Conversely, AraC-binding sites gradually increased repression as they moved further upstream, with a significant inverse relationship between operator distance and expression, though the effect size is small ($P = 2.19 \times 10^{-5}$, ANOVA). To see whether these relationships would change when DNA looping was possible, we evaluated the effect of moving the distal site when a proximal site was also present (Fig. 1g). To directly observe the impact of the distal site, we determined the expression at each distal position relative to expression when only the proximal site was present. Coupled with a proximal site, a majority of tested TFs exhibited different repression patterns as the distal site was moved. For AraC, GalR, and LacI the distal sites reduce expression more with a proximal site present than without (AraC: 1.18-fold, $P = 1.83 \times 10^{-8}$, Welch's two-sided *t* test; GalR: 1.35-fold, $P = 2.82 \times 10^{-11}$, Welch's two-sided *t* test; LacI: 1.37-fold, $P = 4.65 \times 10^{-14}$, Welch's two-sided *t* test). This enhanced repression by distal sites when a proximal site is present indicates the existence of synergistic interactions between these sites. Furthermore, repression by these distal sites followed a 10–11 bp periodicity as they were placed incrementally further from the proximal site, which may indicate the formation of DNA loops at the *lacUV5* promoter. LldR, PurR, and GlpR distal sites did not show significantly enhanced ability to repress when a proximal site was present ($P > 0.4$ in all cases, Welch's two-sided *t* test), indicating these TFs may not participate in looping-based repression. Additionally, distal site and loop-mediated repression differed between TFs tested which is likely due to differences in how these proteins are situated on their binding sites or oligomerize to form DNA loops. Thus, we find different repressor systems exhibit unique relationships between operator spacings and repression, highlighting the need to study these systems individually.

**Tuning binding site strengths alters inducible promoter behavior.** Having identified the optimal spacing for LacI sites at the *lacUV5* architecture, we next sought to learn how these sites may be manipulated to generate *lacUV5* variants with minimal leakiness and maximal fold change, properties that are desirable in synthetic applications. Previously, we found testing large libraries of promoters composed of various combinations of sequence elements allows us to characterize the contribution of individual sequence elements and reveal interactions between them[20,53]. Employing a similar MPRA strategy, we designed and assayed a library of 1600 inducible promoters, referred to as Pcombo, composed of all possible combinations of one of ten proximal LacI-binding sites at +11, four −10 elements, four −35 elements, and ten distal LacI sites at −90 (Fig. 2a). To cover a wide range of expression, we selected −10 and −35 element

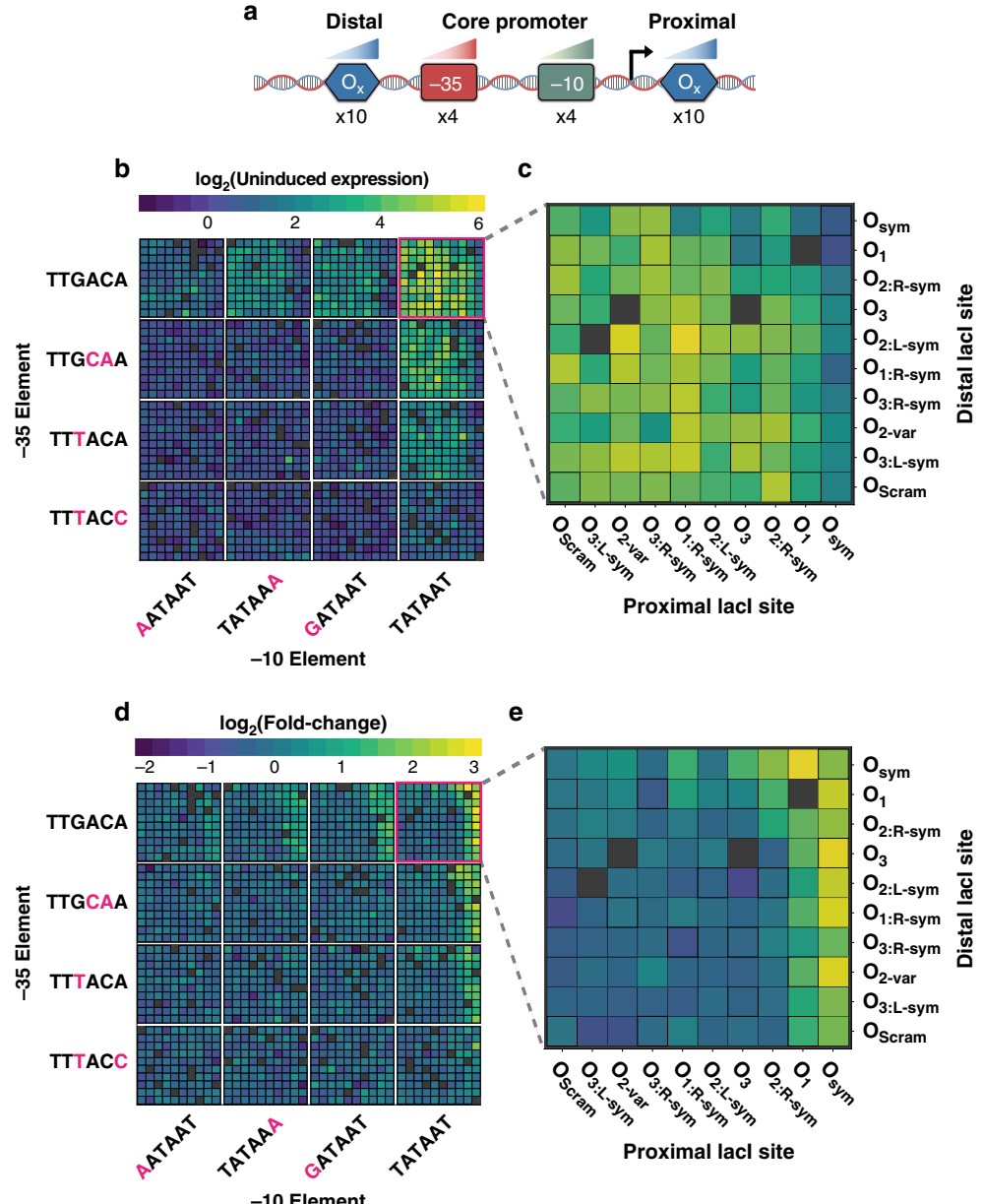

**Fig. 2 Tuning binding site strengths alters inducible promoter behavior. a** Pcombo library schematic consists of all combinations of one of ten proximal LacI-binding sites, four −10 elements, four −35 elements, and ten distal LacI sites. **b** Uninduced MPRA expression for all assayed Pcombo variants. Grid positions for the −10 and −35 motifs are arranged according to median induced expression, from the weakest to consensus sites (−10:TATAAT and −35: TTGACA). Gray boxes indicate sequences that were not measured by the assay. **c** Uninduced expression for assayed Pcombo variants containing consensus σ70-binding sites. **d** Fold change for all assayed Pcombo variants. Fold change is determined by the ratio of MPRA expression at 1 mM IPTG relative to 0 mM IPTG. **e** Fold change for assayed Pcombo variants containing consensus σ70-binding sites. Source data are available in the Source Data file.

variants previously shown to span a range of RNAP-binding affinities[6,20,53]. Similarly, we chose a range of LacI-binding site variants from well-characterized genomic operator sites ($O_1$, $O_3$, $O_{sym}$)[10,18], a variant of the natural $O_2$ site, $O_{2\text{-var}}$, and a series of LacI sites created from different combinations of the monomeric halves of each of these dimeric binding sites (Supplementary Table 2). While $O_1$ is the naturally occurring operator site reported to have the highest affinity for LacI, the synthetic $O_{sym}$ is a symmetrized variant with an even higher affinity[18,54]. Expression data for these variants was collected in both uninduced (0 mM IPTG) and fully induced conditions (1 mM IPTG). We recovered expression measurements for 1493 variants within this

library (93.3%) with an average of 9 barcodes measured per variant. We observed high expression correlation between biological replicates in both the induced and uninduced conditions (Induced: $r = 0.945$, $P < 2.2 \times 10^{-16}$, Uninduced: $r = 0.955$, $P < 2.2 \times 10^{-16}$, two-sided Student's $t$ test) (Supplementary Fig. 2a).

We first explored how the composition of sequence elements determined uninduced expression or leakiness. Library variants exhibited a 267-fold range of uninduced expression; even amongst variants containing the same core promoter σ70 elements, expression varied by up to 96-fold (Fig. 2b). As has been previously reported in comparable promoter variants[20], detectable expression levels were only observed when either the

−10 or −35 elements matched the consensus sequence. In the uninduced state, promoters composed of the consensus −10/−35 elements exhibited the greatest leakiness, with up to 21-fold higher average expression than that of promoters composed of weaker −10/−35 elements. Effective repression generally required a strong LacI operator site, such as $O_{sym}$ and $O_1$, in the proximal position, especially amongst variants with consensus −10/−35 elements (Fig. 2c). Although pairs of repressors exhibited similar effects on different combinations of −10 and −35 elements, there was still variability in these effects which may be due to biological and experimental noise at low levels of expression or interactions between sequence elements[9,20].

We next explored how the interplay between operator sites and RNAP-binding site strengths influences fold change between induced and uninduced states. We determined the fold change of variants by normalizing induced and uninduced measurements to negative controls in each condition and calculating the ratio of normalized induced expression to normalized uninduced expression. Overall, we observed a 40-fold range of fold changes in expression (Fig. 2d). Promoters consisting of the consensus −10 and −35 sites exhibited the highest fold changes; however, these values were highly variable depending on the variant's operator site composition (Fig. 2e). Amongst promoters containing these core sites, we found that operators in the proximal site were largely deterministic of fold change, with promoters containing strong operators ($O_1$ and $O_{sym}$) in the proximal site yielding 4.61-fold higher fold changes on average than promoters containing weak operators in the proximal site ($P = 1.44 \times 10^{-6}$, Welch's two-sided $t$ test). We attribute this to the importance of the downstream operator in blocking RNAP binding and transcriptional initiation[10,55]. As expected, promoters containing $O_{sym}$ in the proximal site generally drove the highest fold change, however, pairing with another $O_{sym}$ in the distal site surprisingly decreased fold change relative to other variants. Notably, while the consensus core promoter containing $O_{sym}$ in both the proximal and distal sites yielded a fold change of 4.63, its counterpart containing the weaker $O_1$ variant in the proximal site drove an increased fold change of 8.97. While the promoter containing $O_{sym}$ in both the proximal and distal sites had 1.77-fold lower uninduced expression compared to its counterpart with a weaker $O_1$ in the proximal site, induced expression was also 3.43-fold lower (Supplementary Fig. 3a). Thus, $O_{sym}$ in both the proximal and distal sites decreased expression in the induced state by a larger magnitude than in the uninduced state, resulting in a lower fold change.

To investigate this unusual phenomenon, we determined which proximal/distal site pair resulted in maximal fold change for other −10/−35 site pairs. Interestingly, we observe that maximal fold-change trends with the strength of the proximal site, but the optimal distal site varies on a core promoter basis. For example, the optimal distal site for promoters containing just one of the consensus −10/−35 sites was the comparably weaker $O_{1:R-sym}$ (Supplementary Fig. 3b), demonstrating that promoter architectures incorporating the strongest repressor binding elements available may not always yield the highest fold-change levels.

**Biophysical modeling of inducible promoter activity**. We set out to clarify the conditions for optimal fold change by combining our experimental measurements with a simple statistical mechanics binding model (described in Supplementary Note). To that end, we modeled promoter architecture by enumerating the various promoter states containing all combinations of RNAP binding, LacI binding, and LacI looping (Supplementary Fig. 4a). We assume that all states where RNAP is bound and the proximal LacI site is not bound to give rise to gene expression $r_{max}$, whereas

all other states have a small background level of gene expression $r_{min}$[9,56]. The relative probability of each state is given by $e^{-\beta E}$ where $E$ equals the sum of all binding free energies arising from binding or looping (Supplementary Fig. 4a). In addition, we include an additional term to scale values when in the presence of IPTG. Using this statistical mechanics model of gene expression, we inferred the binding energies of each promoter element and compared the resulting fits for the 1493 different promoters in the absence of IPTG (Fig. 3a, $r^2 = 0.79$, $P < 2.2 \times 10^{-16}$, two-sided Student's $t$ test, parameter values in Supplementary Fig. 4b). Interestingly, we found that all parameters could be fit using as little as 5% of the library and retain the ability to accurately predict the other 95% of variants when used in this model framework (Supplementary Fig. 5a). Furthermore, this model enables us to extrapolate the gene expression for promoter architectures with arbitrary binding strengths spanning the theoretical parameter space (Fig. 3b).

We then used fit gene expression in the induced and uninduced states to explore how fold change varies as a function of inferred LacI binding energies (Fig. 3c). Returning to our earlier result, we confirmed that pairing together the consensus −35/−10 RNAP-binding site with a proximal and distal $O_{sym}$ LacI site (binding energy $-2.4 k_B T$; Supplementary Fig. 4b) leads to suboptimal fold change. Previously, measurements have shown that even at 1 mM IPTG, a small number of LacI dimers are still active[57], and hence the large binding affinity to $O_{sym}$ sites may drive measurable repression levels[58,59]. Both our experimental measurements and statistical mechanics model support this notion, demonstrating that using $O_{sym}$ at both the proximal and distal sites leads to the sufficiently strong binding that overwhelms the small number of active repressors per cell, leading to reduced gene expression even at 1 mM IPTG (Fig. 3b). Instead, the promoter architecture that maximizes fold change couples the strong −10 and −35 RNAP elements with near-maximal LacI operator site strengths that are sufficiently strong enough to repress in the absence of IPTG but not in the presence of saturating IPTG. We also observed that to achieve optimal induction in weaker promoters, the strength of the LacI operator sites should decrease by a commensurate amount (Fig. 3d).

**Additional operator sites can promote or reduce induction response**. We next sought to explore how these behaviors would change in the context of alternative architectures in which we varied the operator number, placement, and RNAP-binding contacts. Based on our previous characterization of the 1600 Pcombo variants, we speculated whether an additional distal operator site could improve the fold change of promoters. In particular, we expected that an additional distal site would enhance repression, as multiple upstream sites would increase the probability of repressor binding and loop formation. To investigate this, we synthesized and tested 2000 *lacUV5* variants within a library we call Pmultiple. This library resembled Pcombo except for the inclusion of an additional modular LacI-binding site, which we refer to as the "distal+ " site, immediately upstream of the distal binding site. The final design was composed of each combination of five distal+ operator sites, five distal operator sites, four −10 elements, four −35 elements, and five proximal operator sites for a total of 2000 variants (Fig. 4a, top). Using our MPRA, we measured expression for 1638 of these variants (81.9%) in the absence of IPTG and at 1 mM IPTG with an average of 8–9 barcodes measured per variant (Supplementary Fig. 2b). To determine the effect of the distal+ site, we compared the fold change of each Pmultiple variant to Pcombo variants composed of the same distal, −35, −10, and proximal sites. We limited our analysis to studying promoters with consensus core

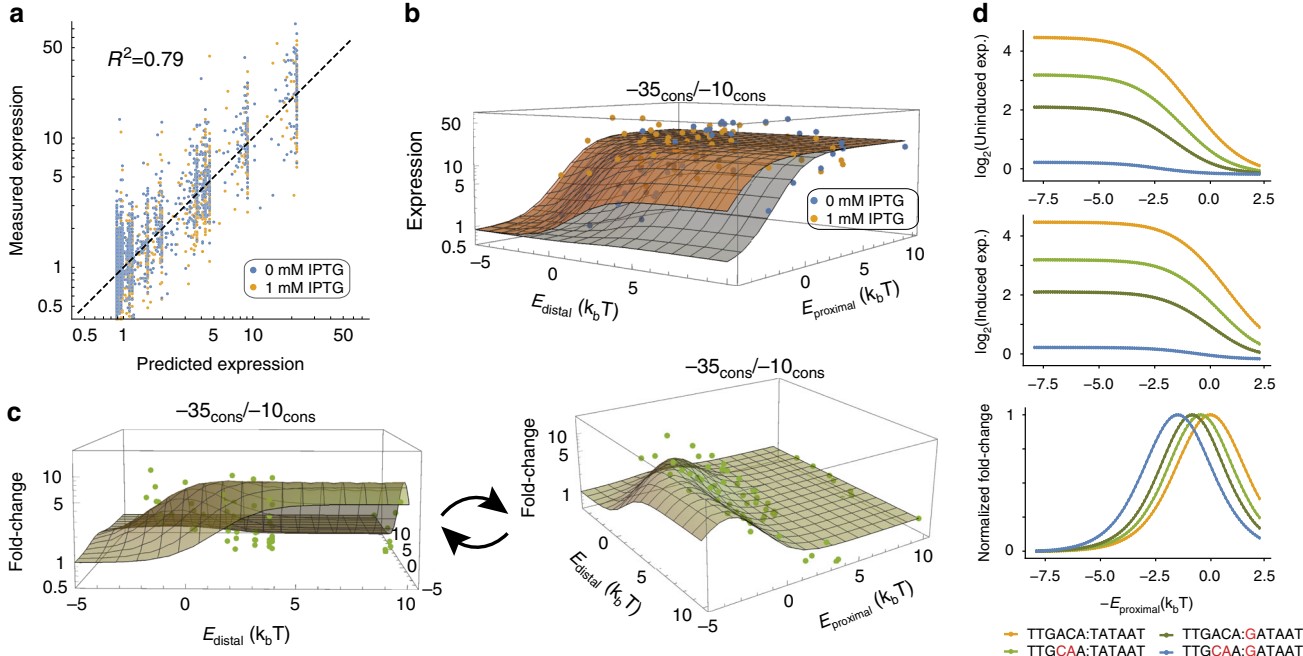

**Fig. 3 Thermodynamic modeling of *lacUV5* promoter architecture. a** Correlation between actual *lacUV5* variant expression and expression fit by our thermodynamic model ($r^2 = 0.79$, $P < 2.2 \times 10^{-16}$, two-sided Student's *t* test). **b** Induced and uninduced gene expression across the distal and proximal site binding energy parameter space. **c** Fold change (FC) in gene expression as a function of distal and proximal binding site energies. In panels **b** and **c**, each dot represents experimental data whereas the grid lines denote the inferred expression of a promoter with the proximal ($E_{proximal}$) and distal ($E_{distal}$) LacI binding energies shown. **d** As promoter strength decreases, optimal induction responses are achieved at lower proximal LacI binding site energies. Promoter binding sites are shown as −35 sequence: −10 sequence. These trends are shown in the context of an $O_1$ distal site ($E_{distal} = -0.23$). Source data are available in the Source Data file.

promoter elements as well as an $O_1$ or $O_{sym}$ proximal site to best capture the repressive effects of the distal+ element. The addition of the distal+ site to the Pcombo architecture spanned a 5.4-fold fold change range, largely determined by both distal and distal+ site identity (Fig. 4a, bottom). We observed that a strong distal+ operator site can consistently compensate for a weak distal operator site to decrease leakiness (Supplementary Fig. 6a) and improve fold change. For example, adding an $O_1$ distal+ site to variants with the weakest distal operator, $O_3$, resulted in a 2.93-fold change. However, when the distal site was already strong, adding a distal+ operator decreased expression fold change. Upon further investigation, we found that in cases where a strong distal site was already present, the addition of a strong distal+ site actually increased leakiness and induced expression of the system, suggesting that the distal+ site may be inhibiting distal site repression of the promoter (Supplementary Fig. 6a, b). Thus, we conclude that additional distal operator sites can improve the fold change of inducible systems by reducing the uninduced expression or have negative effects if they lead to competition with another strong distal site.

Finally, we explored whether our previously established statistical mechanics model could accurately predict the expression of variants in this library. We extended our model framework to account for the different promoter states available to the Pmultiple architecture (described in Supplementary Note) while retaining the same parameter values that fit the Pcombo library. Despite a lack of training on promoters of this architecture, the model was still able to predict the expression of Pmultiple variants with impressive accuracy (Supplementary Fig. 5b, $R^2 = 0.73$, $P < 2.2 \times 10^{-16}$, two-sided Student's *t* test). We expect the drop in accuracy is related to the observed interactions between the distal and distal+ sites, which will require further studies to parameterize. Nonetheless, we show that this adaptable model framework is robust even across previously unseen sequence architectures.

**Changing repression mode alters activity independent of sequence element composition**. Next, we explored how repositioning operator sites influence repression of the *lacUV5* promoter. Previous work indicated operator sites placed within the spacer region, the segment of DNA between the −10 and −35 elements, enabled strong repression[13]. Notably, this positions the operator such that it directly competes with RNAP binding. Furthermore, this architecture is desirable for synthetic applications as it avoids placing operators downstream of the TSS, like the proximal site[14]. To explore this concept in-depth, we synthesized Pspacer, a library of 4400 variants containing all combinations of five distal operator sites, four −35 elements, four −10 elements, and five spacer operator sites (Fig. 4b, top). Because this spacer region is 17 bp and the LacI operators we use are 21 bp, operator sequences were truncated by 2 bp at their termini so as not to overlap the −10 and −35 motifs. In order to determine the optimal spacing between the distal and spacer operator sites, we also tested these combinations with interoperator distances between 46 and 56 bp. We recovered expression data for 3769 (85.7%) of these variants in the absence of IPTG and at 1 mM IPTG with an average of 7 barcodes per variant (Supplementary Fig. 2c). The distance between the spacer and distal operator sites did not appear to significantly affect the fold change of the promoters at the $P < 0.05$ threshold (ANOVA), which may be because some of the tested distances were insufficient to enable the formation of DNA loops[17,37] (Supplementary Fig. 7a, b).

With all operator spacings tested appearing equivalent, we subset our analysis to variants with an interoperator distance of

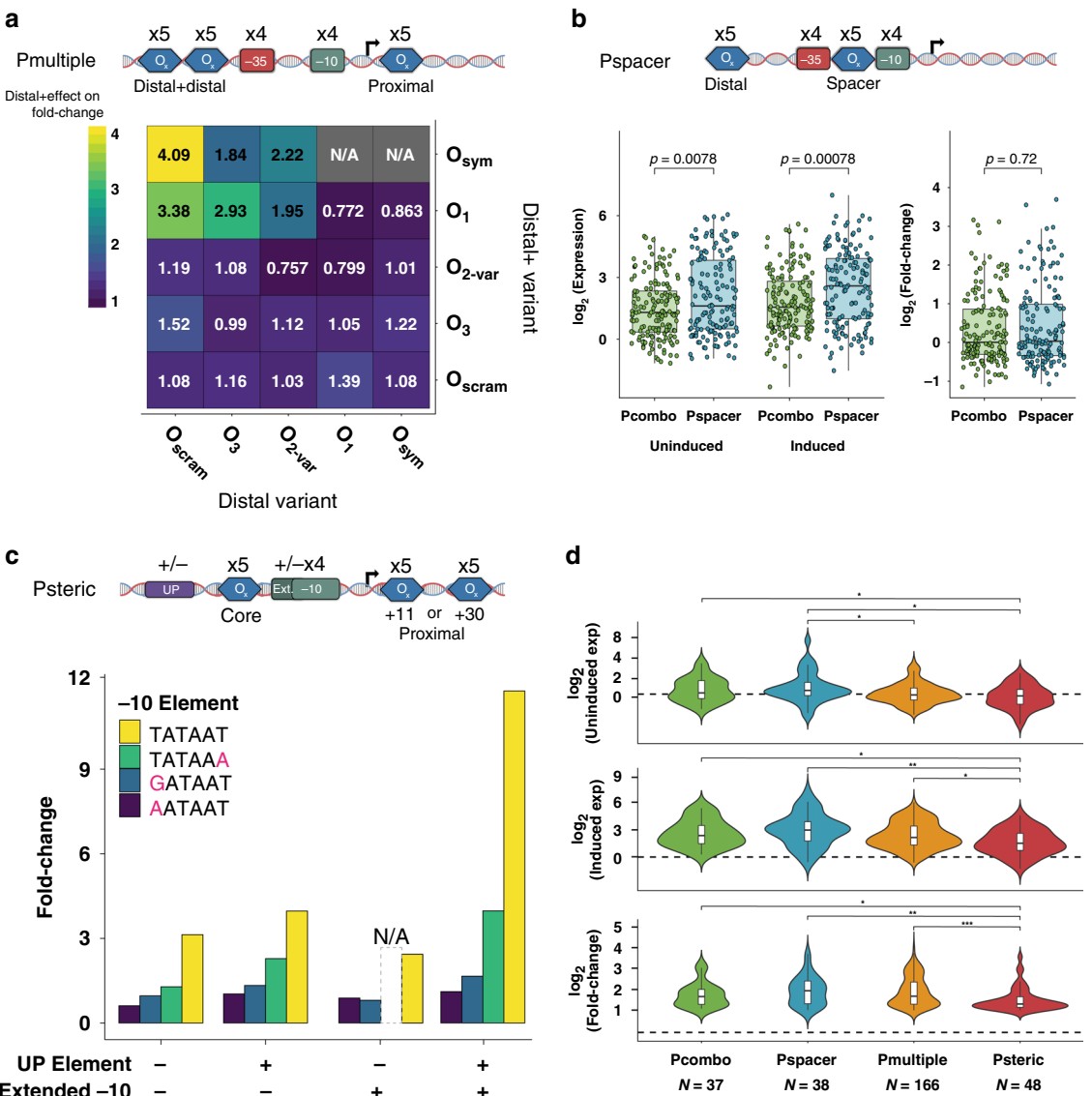

**Fig. 4 Optimizing alternative IPTG-inducible promoter architectures. a** Top: Design for Pmultiple library. Bottom: The average effect of the distal+ site (rows) on fold change given the distal site identity (column). Here, we examine consensus −10/−35 promoters containing $O_1$ or $O_{sym}$ in the proximal site. **b** Top: Design for Pspacer library. Bottom: Comparison of uninduced expression, induced expression, and fold change between variants composed of the same sequence elements in the Pspacer and Pcombo architectures (two-sided Mann–Whitney U tests, n = 305). We examined only active promoters containing a consensus −10 and/or −35 sequence. In each boxplot, the lower, middle, and upper hinges correspond to the first quartile, median, and third quartile, respectively. Whiskers represent 1.5 × IQR from the lower and upper hinges. **c** Top: Design for Psteric library. Bottom: The fold change of promoters containing $O_1$ in both the core and proximal sites and a 56 bp interoperator distance. Here, we examine the effect of the −10 element in conjunction with the strongest UP and extended −10 element combinations. N/A indicates data missing from our analysis. **d** Distributions of uninduced expression, induced expression, and fold change for variants with fold change ≥2 in each library. The dashed line separates active from inactive sequences and is set as the median of the negative controls + 2*median absolute deviation (two-sided Mann–Whitney U tests with Benjamini–Hochberg correction, *P ≤ 0.05, **P ≤ 0.01, ***P ≤ 0.001). The exact P values are: uninduced (Pspacer-Psteric: P = 0.029, Pspacer-Pmultiple: P = 0.040, Pcombo-Psteric: P = 0.045), induced (Pspacer-Psteric: P = 0.002, Pcombo-Psteric: P = 0.013, Pmultiple-Psteric: P = 0.013), fold change (Pmultiple-Psteric: P = 0.0009, Pspacer-Psteric: P = 0.0045, Pcombo-Psteric: P = 0.040). Source data are available in the Source Data file.

55 bp, which is reportedly amenable to looping[37]. Similar to variants with the Pcombo architecture, we only observed strong induced expression with promoters containing −10 and −35 elements resembling the consensus (Supplementary Fig. 7c). To see how this change in architecture altered the performance of these promoters, we compared Pspacer variants to Pcombo promoters composed of the same *cis*-regulatory elements. Surprisingly, promoters with the Pspacer architecture had on average 2.16-fold higher uninduced and 1.93-fold higher induced expression (Fig. 4b, bottom). This may be because fewer repressed

states are possible in this architecture, thereby pushing the system to be more active. Alternatively, this increased expression may be due to greater spacer %AT content within spacer LacI sites which may enhance promoter melting[20,60] (Supplementary Table 3). Despite these higher expression values, Pspacer variants had comparable levels of fold change to corresponding variants of the Pcombo architecture (Fig. 4b, bottom).

**Altering RNAP-binding contacts.** Finally, we tested whether altering RNAP contacts could modify the behavior of inducible

systems. Although all promoters tested thus far were designed to contact RNAP through the σ70 −35 and −10 elements, previous reports have suggested the possibility of engineering promoters lacking −35 elements[61,62]. In these cases, additional compensatory binding sites for transcription factors or RNAP are necessary to recruit RNAP and enable transcription. In addition to the −35 and −10 motifs, RNAP binding may be enhanced by an extended −10 TGn[63,64] motif and an AT-rich UP element[65,66] upstream of the −35 that stabilizes the RNAP α-subunit. However, it is not yet clear if these additional sequences are sufficient to compensate for the lack of a −35 element or how such an architecture would behave in an inducible context.

We synthesized and tested a library of 1600 *lacUV5* variants, called Psteric, containing every combination of four −10 elements, five core operator sites centered at −26 instead of the −35 element, five proximal operator sites, and four UP elements in the presence or absence of an extended −10 motif (Fig. 4c, top). Furthermore, we positioned the proximal operator site centered at either the canonical +11 position or at the +30 position. At +30, the proximal operator is 56 nucleotides away from the core operator, which is near an optimal distance for repression loop formation[37]. We recovered expression data for 1369 of these variants (85.6%) in the absence of IPTG and at 1 mM IPTG with an average of 8 barcodes per variant (Supplementary Fig. 2d). We first examined library variants lacking functional LacI operator sites to identify combinations of −10 elements, extended −10 elements, and UP elements yielding functional promoters. Although weak or no transcription was detected from promoters with only a −10 element, we found the UP element and extended −10 synergistically increased expression, with up to 13-fold greater expression than promoters containing just a consensus −10 (Supplementary Fig. 8a).

Next, we compared two operator placements within this architecture to evaluate whether they enabled inducible behaviors. First, we found variants with the highest fold change were constructed with proximal operator sites located at the +30 position relative to the TSS, though the overall median fold change of promoters did not differ between the two proximal operator site positions (Supplementary Fig. 8b). Second, we found the inducibility of these promoters relies on the presence of a UP element, an extended −10, and a strong −10 motif. When all three are present, promoters containing a proximal operator site located at the +30 position exhibit up to an 11.8-fold response to IPTG (Fig. 4c, bottom). Despite the apparent viability of this architecture, we found that the highest expressing promoters generally contained $O_{scram}$ or $O_1$ core operator sites (Supplementary Fig. 8c). In these cases, we found operator sites tended to partially match the −35 motif, although they were not placed in the optimal position relative to the −10 motif (Supplementary Fig. 8d).

**Comparison of optimized alternative *lacUV5* promoter architectures**. To gauge how our alternative inducible promoter architectures perform relative to one another, we compared the distributions of fold changes between each library. To focus on inducible variants, we limited our analysis to promoters with fold change ≥2. Of the thousands of promoters tested, relatively few were capable of induction, highlighting the difficulty in engineering these systems. Each architecture generated promoters with similarly wide ranges of uninduced expression, induced expression, and fold changes (Fig. 4d and Supplementary Table 8). However, overall comparisons revealed significant differences between these distinct architectures. In particular, Psteric members drove the lowest uninduced and induced expression, likely due to the noncanonical RNAP contacts with these

promoters ($P < 0.05$, two-sided Mann–Whitney $U$ test with Benjamini–Hochberg correction). Variants with the highest fold change were isolated from Pspacer and Pmultiple libraries, highlighting the potential benefits of exploring beyond canonical regulatory architectures. Although previously we found Pspacer variants exhibited greater uninduced and induced expression than Pcombo variants, we did not observe this phenomenon between these subsets of each library.

**Validation of functional inducible variants using a fluorescent reporter**. Finally, we sought to identify inducible variants superior to the canonical *lacUV5* promoter. From all four architectures, we individually evaluated promoter sequences exhibiting higher fold change with low leakiness by using flow cytometry to measure sfGFP expression in uninduced (0 mM IPTG) and fully induced (1 mM IPTG) conditions (Fig. 5). Compared to *lacUV5*, all variants exhibited improved fold change (min: 9.5×, max: 21.0×, *lacUV5*: 4.1×). In particular, a Pmultiple variant demonstrated >5-fold higher fold change than *lacUV5*. Many variants, especially Psteric promoters, exhibited low leakiness while maintaining comparable induced expression. Activity measurements using flow cytometry well-correlated with MPRA measurements (induced: $r = 0.701$, uninduced: $r = 0.981$, fold change: $r = 0.885$) (Supplementary Fig. 9). Lastly, we found that all architectures demonstrated similar input–output relationships as *lacUV5* in response to IPTG induction at variable concentrations (Supplementary Fig. 10).

## Discussion

While current strategies for tuning inducible systems involve arbitrarily manipulating individual operator sites and core promoter elements, these approaches provide little insight into the combinatorial interactions modulating expression. Here, we implemented a MPRA to measure gene expression of nearly 9000 different promoter variants, learning the design logic for multiple sequence architectures. We found different repressors exhibit unique relationships between the operator placement and repression, highlighting the need to study these systems individually. We focused on the canonical P*lacZYA* inducible promoter, finding that induction largely depends on an interplay between the repressor and the core promoter elements. Notably, RNAP and repressors compete for binding, such that promoters containing near-consensus −35 and −10 σ70 elements are functionally irrepressible unless matched with correspondingly strong repressor sites. However, as has been previously shown[56,57], the strongest LacI sites are repressive even in the presence of inducer, reducing fold change. Both a thermodynamic model and our empirical measurements agree that fold change is optimized by selecting repressor binding sites commensurate to the strength of the promoter.

Beyond studying combinatorial effects within the P*lacZYA* architecture, we investigated these interactions in alternative promoter contexts. Characterizing the dynamic range of expression of alternative inducible promoter architectures expands our ability to fine-tune metabolic pathways for generating chemical compounds, especially when products are toxic to the host system[67]. Furthermore, this approach could be applied towards identifying design rules to minimize leakiness and maximize fold change in other bacterial repressor systems that likely operate under similar thermodynamic principles. To our knowledge, a systematic analysis exploring a similar interplay between TF and core promoter strengths in eukaryotic systems has yet to be performed, however many MPRAs have explored the regulatory role of TFs[23–29,68] and core promoter[69] binding site compositions separately. Thus we predict the approach presented here can

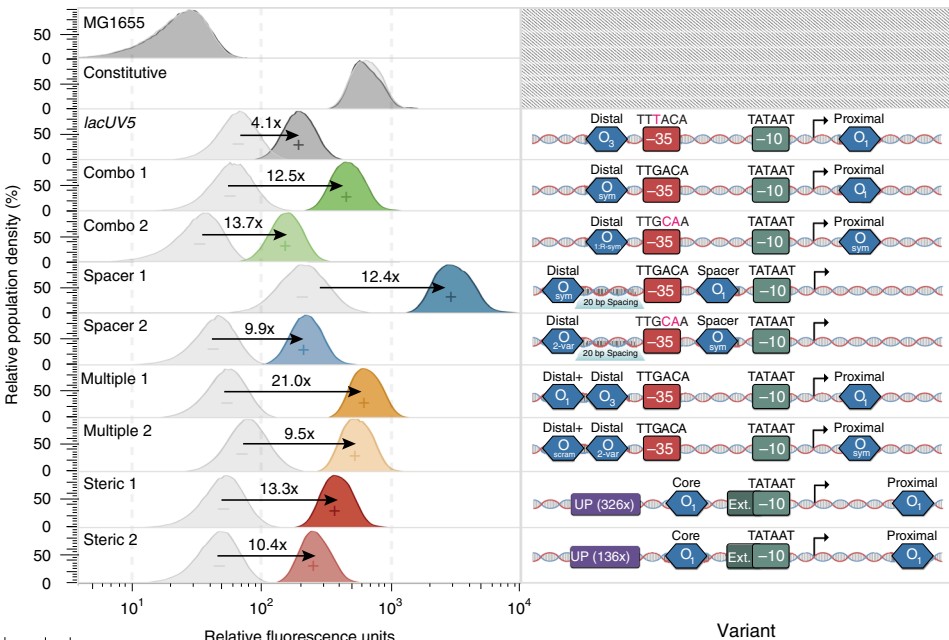

**Fig. 5 Characterization of functional inducible variants using a fluorescent reporter.** Fluorescence measurements of selected variants for induced and uninduced states determined using flow cytometry. Fold change of each variant was estimated after background subtracting induced and uninduced expression. "−" represents the promoter in an uninduced state while "+" represents induction after 1 mM IPTG. Source data are available in the Source Data file.

inform us about the interactions between TF and core promoter sites in other systems.

Ultimately, this systems analysis of inducible promoter regulation demonstrates the utility of combining rational design with large-scale multiplexed assays. Testing sequence libraries in multiplexed formats enabled the exploration of distinct functional designs as well as the discovery of promoter variants with desirable properties. In addition, this assay provides a reliable means for exploring the effects of specific genetic variants, which can reveal insights into promoter mechanisms and sequence–function relationships.

## Methods

**Promoter library design**. A library of 624 variants was created to test the effects of altering the spacing between LacI, AraC, GalR, GlpR, LldR, and PurR operator sites. The core promoter PlacL8-UV5, is the endogenous lacZYA promoter region with L8 and L29 mutations in the CAP site to render it catabolite insensitive (−55 C- > T, −66G- > A) as well as UV5 mutations in the −10 region to increase activity (−9, −8 GT- > AA)[70–72]. Pairs of 23-bp operator sites were acquired from endogenous loci reported by RegulonDB[49] (ver 8.0) (Supplementary Table 1). For sites under 23 bp in length, the surrounding sequence of the native genomic context was included. In all cases, the downstream site found at the endogenous loci, with respect to the regulated promoter orientation, was used as a proximal site in our designs while the upstream sequence was used as the distal site. For each pair of operator sites, a series of variants were designed where the proximal operator was centered at +12 (spanning +1 to +23) and the distal operator varied from positions −83 to −116. Similar series of variants were also designed, in which the sequence of the proximal site or distal site was shuffled to obviate the activity of the operator.

A library (Pcombo) of 1600 lacUV5 variants composed of each combination of 10 proximal operator sites, 10 distal operator sites, four −10 elements, and four −35 elements was designed. The operator sites were selected to span a wide range of lacI-binding affinities (Supplementary Table 2). These consisted of two native LacI operators ($O_1$ and $O_3$) and a variant of the native $O_2$ lac operator with three mutations ($O_{2\text{-var}}$). In addition, $O_{sym}$ and six other synthetic operators ($O_{1:R\text{-sym}}$, $O_{2:L\text{-sym}}$, $O_{2:R\text{-sym}}$, $O_{3:L\text{-sym}}$, $O_{3:R\text{-sym}}$) were used with the latter being designed by creating palindromic sequences based on either the left or right halves of each native sequence. Lastly, a scrambled operator ($O_{scram}$) composed of a random scrambling of the $O_1$ sequence served as a negative control. The −10 and −35 sites were selected to span a range of binding affinities for RNA Polymerase and obtained from a previous characterization[6,8,20] (Supplementary Tables 4–5). Each variant was composed of a combination of these elements placed onto catabolite

insensitive (L8, L29 mutant), lacZYA promoter with the proximal site placed at +11 and the distal site placed at −90, which was found to enable strong looping in the assay of transcription factor spacing.

A library (Pmultiple) of 2000 lacUV5 variants composed of each combination of one of five distal+ operator sites, five distal operator sites, five proximal operator sites, four −10 elements, and four −35 elements was designed. The $O_1$, $O_3$, $O_{2\text{-var}}$, $O_{sym}$, and $O_{scram}$ operators from the Pcombo library were selected as the five operator sites for testing. In addition, the same −10 and −35 elements from the Pcombo library were selected. This library was constructed with sequence elements placed in the same positions as the Pcombo library, with the exception of the distal + sequence being placed immediately upstream of the distal site.

A library (Pspacer) of 4400 lacUV5 variants composed of each combination of five distal operator sites, four −35 elements, four −10 elements, and five spacer operator sites was designed. In order to fit the 17-bp spacer region, two base pairs were trimmed from each end of the spacer operator sites (Supplementary Table 2). The same operators, −10 elements, and −35 elements from the Pmultiple library were selected. Lastly, the distal operator site was tested at 10 different spacings relative to the core promoter, ranging from 20–30 bp from the 5' most end of the −35 element. These 20–30 bp spacings resulted in an interoperator distance of 46–56 bp.

A library (Psteric) of 800 lacUV5 variants composed of each combination of four −10 elements, five core LacI sites centered at −26, five proximal operator sites, and one of four UP elements in the presence or absence of an extended −10 motif was designed. The same operator sites and −10 elements from the Pmultiple library were selected. Proximal operator sites were tested when centered at both the +11 and +30 positions relative to the TSS. The UP elements selected were obtained from a previous characterization[20,73] (Supplementary Table 6). In addition, the extended −10 element TGG was used as this is the most commonly found version of an extended −10[64].

**Library cloning**. The library was synthesized by Agilent and then resuspended in 100 μL of elution buffer before cloning into plasmid pLibacceptorV2 (Addgene ID no. 106250). The transcription factor spacing library was ordered separate from the other libraries, which were altogether synthesized and tested in a multiplexed pool. First, the library was amplified with KAPA SYBR FAST qPCR Master Mix (#KK4600) utilizing primers GU 132 and GU 133 at 10 μM to determine Cq values. Afterward, the library was amplified with NEBNext® Q5® Hot Start HiFi PCR Master Mix (#M0543S) at 11 cycles using primers GU 132 and GU 133 as well, in triplicate. Replicates were pooled, then cleaned with Zymo Clean and Concentrator Kit (#D40140).

To barcode the library, each library was amplified with NEBNext® Q5® Hot Start HiFi PCR Master Mix (#M0543S) for 10 cycles using primers GU 132 and GU 134. Library ends were then digested with SbfI-HF (NEB #R3642S) and XhoI (NEB #R0146S) by incubating at 37 °C for 1.5 h. The plasmid vector, pLibAcceptorV2, was first maxi-prepped with QIAGEN Plasmid Maxi Kit (#12162), concentrated

with a Promega Wizard SV Gel and PCR Clean-up System (#A9281), and digested with SbfI-HF (NEB #R3642S), SalI-HF (NEB #R3138S), and rSAP (NEB #M0371S) for 1.5 h at 37 °C. Insert (library) and vector (pLibAcceptorV2) were ligated using T7 DNA Ligase (NEB #M0318S), incubating at room temperature for 1 h. The plasmid was then transformed into DH5α electrocompetent *E. coli* cells (New England Biolabs C2989K) and plated for 24 h at 30 °C on LB + kanamycin (25 µg/mL) agar plates. These plates were then harvested in 5 mL of LB and 400 × 10⁶ cells (based on OD₆₀₀) were grown overnight in 450 mL LB + kanamycin (25 ug/mL). This plasmid, consisting of the library cloned into pLibacceptorV2, was isolated and concentrated with Zymo Clean and Concentrator Kit (#D40140).

To clone RiboJ::sfGFP into the plasmid, RiboJ::sfGFP was first amplified with NEBNext® Q5® Hot Start HiFi PCR Master Mix (#M0543S) for 25 cycles using primers GU 99 and GU 100 at 10 µM. This amplicon was then digested with BsaI-HF (NEB # R3535) and NcoI-HF (NEB #R3193S) for 1.5 h at 37 °C. pLib was digested with BsaI-HF (NEB # R3535) and NheI (NEB# R3131S). pLib vector was then ligated with the GFP insert using T7 DNA Ligase (NEB #M0318S), incubating at room temperature for 1 h. This plasmid was next transformed into DH5α electrocompetent cells and plated for 24 h of growth at 30 °C as well, yielding pLib_sfGFP plasmid after maxi-prep.

**Library integration.** The pLib_sfGFP plasmid was first digested with SalI-HF (NEB #R3138S) and NheI (NEB# R3131S) to remove the background. This was then transformed into the landing pad strain, an engineered[20] *E. coli* MG1655 derivative (Yale Coli Genetic Stock Center no. 6300), and grown overnight for 24 h at 30 °C. The following day, plates were scraped and 800 million cells in 200 mL of LB + kan (25 µg/mL) were inoculated overnight at 30 °C.

For library integration, glycerol stocks of landing pad strain with the integration plasmid were grown overnight in 200 mL + kan (25 µg/mL) at 30 °C. 200 million cells from this overnight culture was inoculated the next day into 250 mL LB + 0.2% arabinose + 25 µg/ml Kan at 30 °C for 24 h to induce recombination. The following day, 800 million cells of induced overnight were inoculated into 80 mL LB + 25 µg/mL Kan at 42 °C for heat cure. This was grown to log phase (OD 0.3–0.7) for about 1.5 h. In total, 200 million cells from this log phase culture were plated at 42 °C for 16 h in undiluted, 10⁻⁵, and 10⁻⁶ dilutions. Plates grown overnight were then scraped, and 400 million cells inoculated into 200 mL LB + Kan 25 µg/mL for overnight growth at 37 °C. Ultimately, this was plated again at 30 °C to validate integration (GFP instead of mCherry) and then glycerol stocked after colony PCR for further confirmation.

**Barcode mapping.** The promoter and barcode region from pLib was prepared for sequencing and downstream mapping of the barcodes to their respective variants. Two PCRs were performed to prepare pLib samples for sequencing, the first of which adds sites for the sequencing primer whereas the second PCR adds the adaptors for Illumina sequencing and a unique index DNA label. Each barcode mapping was performed in duplicate.

For the first PCR, the library was amplified with KAPA SYBR FAST qPCR Master Mix (#KK4600) with primers GU 60 and GU 79 at 5 µM to determine Cq values. Afterward, the library was amplified with NEBNext® Q5® Hot Start HiFi PCR Master Mix (#M0543S) at 11 cycles using primers GU 60 and GU 79 at 5 µM as well in triplicate. Replicates were pooled, then cleaned with Zymo Clean and Concentrator Kit (#D40140), eluting into 10 µL of Ultra-pure H₂O.

For the second PCR, Illumina adapters P7, P5, and a unique DNA index were added. The product from the first PCR was amplified with primers GU 70 and GU 86 at 5 µM to determine Cq values. Afterward, the library was amplified with NEBNext® Q5® Hot Start HiFi PCR Master Mix (#M0543S) at ten cycles using primers GU 70 and GU 86 at 5 µM. Since different primers add different indices to each sample, we re-ran the second PCR with a different set of primers to serve as redundancy and allow us to compare sequencing replicates. This process was repeated in a separate PCR, with primers GU 70 and GU 87 also at 5 µM.

Ultimately, each technical replicate was performed in duplicate, cleaned with Zymo Clean and Concentrator Kit (#D40140), and ran on a 1.0% agarose gel for final confirmation. After quality assessment, samples were sequenced on an Illumina Nextseq 500 using a Paired-end 300-cycle kit (2 × 150 bp). Barcodes were mapped to their respective promoter variants using the pipeline from Urtecho et al.[20]. In brief, paired-end reads are merged using PEAR[74] (version 0.9.1). We then extract the first 150 bp of each read, which encodes the promoter variant, as well as the last 20 bp encoding the barcode, and generate a list of barcode-variant associations. Finally, we perform additional filtering steps for quality control purposes.

**Library growth and sequencing preparation.** Library pellets were prepared in both Induced and Uninduced conditions. First, glycerol stocks were inoculated in 100 mL of MOPS with 0.2% glucose + kanamycin (25 µg/mL) at 30 °C for 16 h overnight. The following day, the overnight culture was diluted to OD 0.0005, inoculated into 200 mL MOPS + kanamycin (25 µg/mL) with 0.2% glucose, and grown at 37 °C to OD 0.5–0.55 (~5 h) both with 1 mM IPTG and without.

To harvest RNA pellets, the culture was first cooled for two minutes in an ice slurry while periodically swirling. For each sample, three 50 mL aliquots of culture were poured into pre-chilled tubes and spun for two minutes at 13,000 × g at 4 °C.

The supernatant was poured off. RNA was extracted from *E. coli* pellets using Qiagen RNEasy Midiprep kit (#75142). We performed technical replicates of this extraction (separate RNA extractions of the same culture) with the operator spacing library and biological replicates (Different cultures grown in parallel before separately extracting). Subsequent wash steps concentrated isolated RNA with Qiagen Minelute Cleanup Kit (#74204). Next, isolated RNA was converted to cDNA with Thermo Fisher SuperScript IV (#18090010) following the manufacturer's directions.

To harvest gDNA pellets, 5 mL samples of each culture were then spun down for four min @ 5000 × g. The supernatant was then poured out. DNA from each pellet was then isolated with Zymo Research ZR Plasmid Miniprep Kit (#D4015) for use as normalization.

The barcoded cDNA was amplified with NEBNext® Q5® Hot Start HiFi PCR Master Mix (#M0543S) from 1 µg of gDNA for 14 cycles with primers GU 59 and GU 60 at 5 µM. The product was cleaned with Zymo Clean and Concentrator Kit (#D40140). In all, 1 ng of this sample was amplified again for ten cycles with primers GU 65–68 and GU 70 for indexing, yielding 8 total samples; technical replicates for induced and uninduced cDNA, and induced and uninduced gDNA. Both prepared DNA and RNA library samples were quantified with Agilent Tapestation, then sent for sequencing on HiSeq2500 (SE 50-cycle) to the *Broad Stem Cell Research Center at UCLA*.

A comprehensive list of all primers used in this paper can be found in Supplementary Table 7.

**Data processing.** Following RNA-Seq and DNA-Seq of the barcodes, we quantify the relative abundance of each barcode. Demultiplexed RNA and DNA reads for each biological replicate were converted to counts of each barcode via a custom UNIX script that extracts barcode sequences from individual reads and counts the number of observed reads for each barcode. These barcode counts were normalized using the following formula:

$$\text{Normalized read counts} = \frac{\text{barcode read counts}}{\text{total sample reads}} \times 10^6.$$

Normalized read counts were then merged by common barcode to yield a comprehensive data frame containing normalized read counts for each barcode in each replicate. This data frame was then merged with the barcode mapping data to map normalized read counts to their corresponding promoter. Multiple barcodes could map to a single promoter, thereby providing replicability, and any promoter that contained fewer than three barcodes in any sample was removed. After this filtering step, promoter expression for each replicate was calculated using the following formula:

$$\text{Promoter expression} = \frac{\Sigma(\text{RNA counts for all promoter barcodes})}{\Sigma(\text{DNA counts for all promoter barcodes})}.$$

To normalize promoter expression between induced and uninduced samples, the expression of each promoter was normalized to the median negative control promoter expression in its respective biological replicate. Lastly, the mean expression of the biological replicates was calculated to obtain final expression values for the induced and uninduced conditions.

**Thermodynamic model of gene expression.** For the Pcombo library, initial guesses for the binding energies of each LacI operator site were used as inputs and refined when fitting a statistical mechanics model to the Pcombo promoter expression data. The coefficient of determination ($r^2$) between fit and actual gene expression values was calculated using log₁₀-transformed values to reduce the effects of large expression outliers.

**Individual promoter variant cloning.** Two promoters were selected from each of the libraries, yielding eight total promoters in addition to two controls (a constitutive promoter and UV5). Individual promoter variants were selected from our library of variants based on the highest fold change (induced over uninduced expression) and fold change:noise ratio (fold change over uninduced expression). These sequences were ordered from IDT as gBlocks® Gene Fragments. Full RiboJ: sfGFP was PCR isolated from the original library. Since promoters were to be measured individually, we did not include a barcode in synthesis. Plasmid vector, pLibacceptorV2 was linearized with SbfI-HF (NEB #R3642S) and SalI-HF (NEB #R3138S).

After synthesis by IDT, promoters were amplified using primers GU 142, GU89, and NEBNext® Q5® Hot Start HiFi PCR Master Mix (#M0543S). Each reporter was assembled with Gibson Assembly® Master Mix (NEB #E2611S) using 30 bp overlaps between the plasmid pLibAcceptorV2, the promoter, and RiboJ:sfGFP. Each assembled reporter was separately transformed into *E. coli* DH5α Chemically Competent *E. coli* (NEB #C2987H) yielding 10 total transformed *E. coli* strains containing their respective promoter, RiboJ:sfGFP, and Kanamycin antibiotic resistance. Afterward, the promoter and downstream GFP segment were sequenced from isolated colonies using the same set of primers, GU 142 and GU89, to confirm correct constructs. All products were cleaned with Zymo Clean and Concentrator Kit (#D40140) except for pLibAcceptorV2, which was cleaned with Promega Wizard SV Gel and PCR Clean-up System (#A9281) after DNA isolation with QIAGEN Plasmid Maxi Kit (#12162).

**Individual promoter variant integration**. *E. coli* strains containing library members were grown overnight for 16 h in 5 mL of Luria Broth and kanamycin (25 mg/μL). Afterward, the plasmid was isolated using Zymo ZR Plasmid Miniprep Kit (#D4054) formed into an electrocompetent MG1655 containing an engineered landing pad within the *essQ-cspB* intergenic locus[20] and plated on LB and kanamycin (25 μg/mL) at 30 °C. Two colonies per promoter were resuspended in LB, and inoculated into 5 mL of LB + kanamycin (25 μg/mL) for overnight growth.

Each promoter was separately integrated into the *essQ-cspB* locus using Cre-Lox-mediated cassette exchange. Following overnight growth, cells of this culture were inoculated into 5 mL of LB, kanamycin (25 μg/mL), and 0.2% arabinose (g/mL) and grown for 24 h to induce integration of the reporter cassette. After integration of the reporter cassette through the arabinose-induced Cre system, the residual plasmid was removed through heat-curing. In total, 200 million cells were inoculated into 3 mL of LB and kanamycin (25 μg/mL) and grown at 42 °C for about 1.5 h to reach log phase (OD 0.3–0.7). After this growth, cells were diluted to $10^{-4}$ and plated on LB + kanamycin (25 ug/mL) plates overnight at 42 °C to complete the heat-curing process.

**Flow cytometry**. Glycerol stocks for each promoter were first scraped and inoculated into liquid cultures containing MOPS EZ-Rich Media (TEKNOVA #M2105) and 25 μg/mL of kanamycin at 30 °C for overnight growth. The following day, cells grown overnight were diluted to an OD of 0.002 in MOPS EZ-Rich Media (TEKNOVA #M2105) with 0.2% glucose (g/mL) and 25 ug/mL of kanamycin at 30 °C. These cells were then transferred to 100-mL flasks all containing 15 mL of MOPS EZ-rich media + 0.2% glucose. 1 mM IPTG + 25 μg/mL kanamycin were added to the "Induced" cultures, whereas 25 μg/mL kanamycin was added to the "Uninduced" cultures. These cultures were then grown at 37 °C for 3.5 h. In all, 5 mL of each sample was spun down, the supernatant was decanted, and the cell pellets were resuspended in 1 mL PBS (GIBCO® PBS Phosphate-Buffered Saline 10010023). In total, 1 mL of each sample was filtered into a Falcon 5 mL Polystyrene Round-Bottom Tube with Cell-Strainer Cap. *E. coli* MG1655 was used as a negative control for GFP expression while a constitutively active library member was used as positive. Data was collected using a BioRad S3 Cell Sorter with ProSort Version 1.6 and analyzed in FlowJo (version 10.0.8r1). *E. coli* cells were isolated by gating using FSC and SSC (Supplementary Fig. 11). Fold change was calculated by dividing the median GFP fluorescence of the induced samples by the median fluorescence of the induced samples

**Plate reader assay**. Glycerol stocks for each promoter were scraped and inoculated into liquid cultures containing MOPS EZ-Rich Media (TEKNOVA #M2105) and 25 μg/mL of kanamycin at 30 °C for overnight growth in 5 mL disposable culture tubes. The following day, each promoter was diluted to OD 0.005 in 500 μL of MOPS EZ-Rich Media (TEKNOVA #M2105) with 0.2% glucose (g/mL) and 25 μg/mL of kanamycin and set up for plate reader analysis in triplicates across an IPTG gradient: 0, 0.001, 0.005, 0.01, 0.1, 1 mM. After samples were grown for five hours at 37 °C, 100 μL aliquots were transferred into 96-well flat-bottom microplates. Measurements were taken for wavelengths 650 nm (measures OD) and 520 nm (measures GFP) on the Tecan Infinite M1000 Pro No. 30064852 plate reader. Data were analyzed in Excel (Version 16.41) with the four reads per time point per well averaged and divided by the OD measurement to calculate the GFP fluorescence.

**Reporting summary**. Further information on research design is available in the Nature Research Reporting Summary linked to this article.

## Data availability

Raw data and promoter expression datasets are available without restrictions through NCBI Gene Expression Omnibus (Accession no. GSE145630). All other relevant data are available from the authors upon reasonable request. Source data are provided with this paper.

## Code availability

The Mathematica notebook used for the thermodynamic model, as well as all code for recreating plots, are available at https://github.com/timcyu/inducible_architecture[75]. Statistical significance is reported to a lower limit of $P < 2.2 \times 10^{-16}$, the lowest reportable value by R.

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

## Acknowledgements

This work was supported by the National Science Foundation Graduate Research Fellowship 2015210106 to G.U., National Institutes of Health New Innovator Award DP2GM114829 to S.K., Searle Scholars Program to S.K., U.S. Department of Energy (DE-FC02-02ER63421 to S.K.), UCLA, and Linda and Fred Wudl. We thank the UCLA BSCRC high-throughput sequencing core and Technology Center for Genomics and Bioinformatics for technical assistance; All past and present members of the Kosuri lab for technical feedback; Suzannah Beeler for thoughtful discussions; and Reid C. Johnson for the paper feedback. Lastly, we thank the UCLA Molecular Biology Interdepartmental Graduate Program and UCLA Bioinformatics Interdepartmental Graduate Program.

## Author contributions

T.C.Y., G.U., W.L.L., J.E.D., J.S., G.B., T.E., and S.K. designed the study. T.C.Y. and K.D.I. generated the sequence libraries. T.C.Y., M.S.B., W.L.L., J.S., and G.B. performed the experiments. T.C.Y., G.U., J.E.D., and T.E. analyzed the data. W.L.L. designed the figures. T.E. and R.P. developed the statistical mechanics model. T.C.Y., G.U., W.L.L., J.E.D., J.S., G.B., and T.E. wrote the paper. All authors edited and approved the paper.

## Competing interests

The authors declare no competing interests.
