## [Peer Review File · Nature Communications]

Reviewers' Comments:

Reviewer #1:

Remarks to the Author:

The manuscript authored by Yu et al. presents results from an oligo library SORT-seq experiment carried out in *E. coli* on various promoter architectures. This is an increasingly informative approach to decipher cis-regulatory logic. However, at this point this paper is not ready for publication, as it seems that the authors did not fully mine their data. With better analysis and perhaps machine learning, further insight into bacterial cis-regulatory logic may be achieved leading to a more coherent story.

Major points:

1. The paper is written as collection of experimental data, which seem at time disjointed, rather than a coherent story. Even though pieces of the data are interesting, the authors often fail to elaborate on some interesting findings making for a disjointed presentation. For example, the protein scan (Figure 1) without diving into the implications of the findings (see comments below for details). Next, data mutations of -10 and -35 regions are presented with little discussion of the effects of the mutations. For instance, it does not seem that the same pattern of binding site pairs emerges for different -35 mutations. There is insufficient discussion of this.
2. Figure 1. The authors mention in passing the oscillatory behavior and up-regulation observed for 5 of 6 proteins without a proximal site present. However, they do not at all follow this thread, which is by far the most interesting finding in the paper. There are many questions that emerge from Figure 1 and are completely unaddressed. How do these classic repressors up-regulate expression from -100 bp upstream of the promoter? Where does the helical periodicity dependence come from if there is no looping? As for the looping, both AraC and LacI (the two known looping proteins) behave differently than the rest of the proteins. LacI changes phase by 90° in the oscillation with the proximal site added, while AraC exhibits oscillations. Thus, the rest of the proteins are not likely to loop to the proximal site. Therefore, what is the mechanism of regulation from the distal site? The thermodynamic model that the authors spend a lot of time does not provide an answer. I would suggest that the authors either remove this part, or expand it to provide some mechanistic insight with another OL. As is, these questions are left open and not addressed anywhere else in the paper.
3. Thermodynamic model (TM). The thermodynamic model presented is an incremental progress over what has been published in the past especially by one of the authors (RP) who is the leading expert in this field. In addition, I do not find that the model adds a lot to the story in the paper, it does not provide a novel or predictive insight, and it is not used for predicting optimal variants for verifications. A better model would incorporate the TM approach with the experimental data to train a machine to provide verifiable predictions for an optimal promoter. Ultimately, the field of synthetic biology does not need more partially characterized parts, but rather reliable part or circuit design algorithms, and this modelling approach does not go far enough towards this goal. In order to make a more coherent story and provide genuine progress for the field, such a model needs to be presented - even if its success rate in predicting "optimal" promoters is low.

Reviewer #2:

Remarks to the Author:

This article presents the employment of a rationally designed approach to study the effect of both RNA polymerase binding and LacI elements in various compositions and architectures within IPTG induced promoters in bacteria. The authors utilized an MPRA workflow in order to allow the parallel examination of over 8,200 combinatorial options. The authors performed a thorough characterization of both ends of promoter activity, i.e. repression and transcription, and utilize their

high-throughput data to build a model predicting in silico the best possible outcome – the strongest possible minimal leaky IPTG inducible promoter. Later they construct libraries with various changes to the basic architecture in order to tackle their role in driving minimal baseline expression and the highest possible fold-change (induction). Overall, the paper is well written, easy to follow the scientific logic and could be a good fit for Nature Communications after revision. I have the following comments:

Major points

-In the introduction and discussion, would be good to mention and discuss what has already been done in this space for eukaryotic regulatory elements and how it compares to this work. There are several relevant articles on this that the authors do not mention. Some of these that come to mind are: PMIDs 31792407, 23921661, 2389260, 22609971, 31253799.

-It would be good if the authors discuss how they think their results implicate other E. coli or bacterial promoters. Are they universal or do they think there will be different rules governing function there. How do they compare to eukaryotic ones, i.e. the above examples. Also, how they could be further used in research or industry for example, for the generation of compounds etc.

Minor points

-For the first results paragraph, 'The lacZYA promoter...', Would be good to have a main figure showing everything mentioned there so easy for reader to understand the structure/function of this promoter.

-In the first three results segments, most of the P values are 2.2×10^{-16} , assuming likely that this is the lowest you can go, but need to explain why.

-Line 124 – The authors spent two paragraphs establishing the mechanism of operation of the distal and proximal LacI binding sites, and then just jumped to LacUV5 promoter. Might be due to a gap in my knowledge of E. Coli operons, but I believe a connection should be made between segments

-Line 146-147 – the number format on the p value is wrong, replace $p < 2.2 \times 10^{16}$ to $p < 2.2 \times 10^{-16}$, if the corrected version is what the authors meant.

-Line 148-153 – This entire segment that relates to Figure 1d seems like an experiment that was not performed as an MRPA. If so, please correct the text to represent this. Additionally, add significance stats to the text.

-Lines 159-160 – Authors claims "uniformity of response", it might be beneficial to be more specific and state "uniformity of cyclic behavior".

-Lines 159-165 – If not because of DNA looping, please mention briefly an explanation to the alternating behavior while lacking the proximal site, meaning just based on the distance from the TSS.

-Figure 1c – Axis names, add units –

-Figure 1d – Clear from the text and the legend but for a better visual representation worth adding another bar in black to represent the NULL proximal reference point. Also, lacking error-bars, significance scores and n (if different from the already stated $n=2$ for the MRPA, Figure 1c). One more point, the illustration exhibit the increment movement of the distal site, which does not seem to be part of the this experiment; correct illustration in accordance.

-Figure 1e-f – representation of the results with an increasing distance going left-to-right, while the illustration shows the distal element going right-to-left might be confusing. Consider changing the x-axis to either flipped order to indicate the location of the distal element to match the illustration.

-Figure 1e-f – if possible to add visual representation of when the proximal and distal sites are in phase or out of phase (a shade or faint borders).

-Line 216 – Table S2 is mentioned prior to Table S1 (in material and methods), switch if needed based on final paper outline.

-Lines 217-218 – unnecessary line/paragraph break

-Line 259-260 – be consistent of formatting the fold change numbers either #-fold, just the # or #x.

-Line 324 – be consistent with the format of R2. Either $R2=0.79$ or $R2=.79$

-Line 353-355 – why the authors choose to include another upstream site of the repressor? Was that indicated from their model? If so they should explain this gap in knowledge.

-Line 355 and Figure 4A – from the text I would expect to see a larger result matrix to include 5x distal and 5x proximal and the combination with 4x -10 and -35 sites. The matrix presents only the proximal-distal 25 combinations. Either reflect correctly in figure or revise the text.

-Figure 4C + lines 433-435 – Authors indicate that they changed the location of the proximal site from +11 to +30. This is not indicated in the illustration.

-Figure 4C – the UP element(-) AND Extended -10 (+) group is missing a bar (TATAAA)

-Lines 448-449 – Authors mention highest variants (from Figure S7B), I would add a note that overall the median FC for all variants was not significantly shifted. Additionally, I think a violin plot will work better for Figure S7B.

-Lines 450-452 – It is not clear rather the authors are related to the +30 Proximal site highest FC promoters, or all promoters (+11 and +30). Please clarify in text

-Lines 466-468 – The authors did not relate to the Pcombo vs. Pspacer similar uninduced expression. Overall from Figure 4D it seems that either pcombo or Pspacer behave similarly. Adding a note on that in the text will be good.

-Line 476 – change “four libraries” to “four architectures”

-Figure S9 – It would be better if all y-axis be similar.

We are grateful for the careful and thoughtful feedback of the reviewers and have made substantial changes to the manuscript considering their suggestions. Most notably, we have: 1) Expanded our analysis of how different repressors influence promoter activity as a function of their position within the promoter 2) Performed a new analysis using our thermodynamic model to show how optimal repressor binding site strength changes depending on the strength of the core promoter, and 3) Streamlined the narrative of this work to be more focused on the central question of how promoter binding site parameters (e.g. position, strength, interactions) affect promoter dynamics. We believe that by implementing these changes, as well as the other excellent suggestions of the reviewers, the findings of this work have been made more impactful, convincing, and accessible to a wider audience.

In our response here, we have left the reviewer comments italicized and our responses in bold font. In the cases where the reviewers ask for additional comments or new discussions within the text, we have included these sections in plain text. Lastly, we have indicated changes within sentences as bold and italicized.

Reviewer #1 (Remarks to the Author):

The manuscript authored by Yu et al. presents results from an oligo library SORT-seq experiment carried out in E.coli on various promoter architectures. This is an increasingly informative approach to decipher cis-regulatory logic. However, at this point this paper is not ready for publication, as it seems that the authors did not fully mine their data. With better analysis and perhaps machine learning, further insight into bacterial cis-regulatory logic may be achieved leading to a more coherent story.

Major points:

1. The paper is written as collection of experimental data, which seem at time disjointed, rather than a coherent story. Even though pieces of the data are interesting, the authors often fail to elaborate on some interesting findings making for a disjointed presentation. For example, the protein scan (Figure 1) without diving into the implications of the findings (see comments below for details).

We thank Reviewer 1 for bringing this issue to our attention. In the revisions we have sought to make a more cohesive story on characterizing the sequence parameters and interactions shaping inducible promoter architecture. We have altered the transitions between sections to unify the different analyses and tailored our modeling section to focus on general principles of binding site effects rather than coming across as another tool for prediction.

Next, data mutations of -10 and -35 regions are presented with little discussion of the effects of the mutations. For instance, it does not seem that the same pattern of binding site pairs emerges for different -35 mutations. There is insufficient discussion of this.

Thank you for pointing out the variability of repressor effects across different combinations of -10 and -35s, which we now acknowledge in the text:

Although pairs of repressors exhibited similar effects on different combinations of -10 and -35 elements, there was still variability in these effects which may be due to biological and experimental noise at low levels of expression or interactions between sequence elements^{9,20}.

In addition, because biological measurements are noisy, we have used the statistical mechanics model which has been fit to our data to better present the trends across different -10 and -35 combinations with a new figure (Figure 3D). We have performed an additional analysis to show how optimal repressor binding energies change with the strength of the promoter. Specifically, we use our thermodynamic model to determine induction levels across *proximal* LacI binding site energies for four promoters of varying strength. This analysis demonstrates that optimal induction for weaker promoters is achieved at weaker *proximal* repressor binding energies.

2. Figure 1. The authors mention in passing the oscillatory behavior and up-regulation observed for 5 of 6 proteins without a proximal site present. However, they do not at all follow this thread, which is by far the most interesting finding in the paper. There are many questions that emerge from Figure 1 and are completely unaddressed. How do these classic repressors up-regulate expression from -100 bp upstream of the promoter? Where does the helical periodicity dependence come from if there is no looping? As for the looping, both AraC and LacI (the two known looping proteins) behave differently than the rest of the proteins. LacI changes phase by 90° in the oscillation with the proximal site added, while AraC exhibits oscillations. Thus, the rest of the proteins are not likely to loop to the proximal site. Therefore, what is the mechanism of regulation from the distal site? The thermodynamic model that the authors spend a lot of time does not provide an answer. I would suggest that the authors either remove this part, or expand it to provide some mechanistic insight with another OL. As is, these questions are left open and not addressed anywhere else in the paper.

We thank Reviewer 1 for highlighting this section and motivating us to further investigate these interesting observations. We now investigate the nature of these oscillations and how they relate to the helical phasing between *proximal* and *distal* operator sites. Specifically, we identified the distance between operators where the center of the *distal* site is in phase with the +11 *proximal* sites and by extension the +1 transcription start site (see grey bars below). This shows us interesting differences between the behaviors and mechanism of action of these transcription factor binding sites.

Notably, in the absence of a proximal site, GalR and PurR *distal* sites repress transcription when in phase (grey bars) with the +1 position yet increase expression when out of phase (Figure 1F). On the other hand, GlpR *distal* sites result in activation when in phase with the +1 position and repress when out of phase.

We have expanded the analysis on TF expression oscillation in the main text, reproduced below:

- First, we looked at the effect of moving the *distal* site across 33 nucleotides in the absence of a functional *proximal* site (Figure 1F). We observed a uniformity of cyclic behavior across most repressors tested, suggesting position-dependent effects are a general phenomenon of many TFs. Lone GalR, GlpR, PurR *distal* sites alternated between activation and repression, a phenomenon which has been observed in similar translocations of a LacI binding site upstream of a promoter in the absence of inducer³⁷. ***This may be due to helical positioning of the repressor relative to RNA polymerase and resulting steric interference, or binding-induced DNA distortions***⁵³. Interestingly, we observed TFs exhibiting opposing position-dependent behaviors, where GalR and PurR repressed when the center of the binding sites were approximately in-phase with the +1 TSS position and activated when out of phase. We observed the opposite effect with GlpR sites.

In addition, we include this statement to acknowledge the differences in loop-dependent activities and propose hypotheses for further investigation:

- Additionally, *distal* site and loop-mediated repression differed between TFs tested which is likely due to differences in how these proteins are situated on their binding sites or oligomerize to form DNA loops.

While we think these findings warrant further investigation, especially to learn the mechanisms by which *distal* transcription factors influence expression, we feel this would be beyond the scope of the paper, which intends to learn the relationship between DNA sequence structures and promoter expression.

3. Thermodynamic model (TM). The thermodynamic model presented is an incremental progress over what has been published in the past especially by one of the authors (RP) who is the leading expert in this field. In addition, I do not find that the model adds a lot to the story in the paper, it does not provide a novel or predictive insight, and it is not used for predicting optimal variants for verifications. A better model would incorporate the TM approach with the experimental data to train a machine to provide verifiable predictions for an optimal promoter. Ultimately, the field of synthetic biology does not need more partially characterized parts, but rather reliable part or circuit design algorithms, and this modelling approach does not go far enough towards this goal. In order to make a more coherent story and provide genuine progress for the field, such a model needs to be presented - even if its success rate in predicting "optimal" promoters is low.

We appreciate the reviewer's feedback on our thermodynamic model. We are proponents of using machine learning to predict gene expression, and indeed our previous work [Urtecho 2018] implemented just such an approach. Given the large array of promoter architectures we characterized, we felt that implementing a separate machine learning algorithm in each case would have marginal utility, whereas future work implementing one general algorithm (incorporating the many parameters controlling promoter behavior) accommodating all architectures would merit a project all of its own, and hence we believe it is beyond the scope of this work. Instead, we utilized modeling to support our understanding by targeting a very specific question, namely, why maximum gene expression may not necessarily be achieved by using the strongest repressor and RNAP binding site combinations. That said, we agree that this modeling was not intended to be one of the main results of the paper, and as such, we have cut down our discussion of the model to its main results in two brief paragraphs (one to explain the model and a second to discuss why gene expression decreases when both the proximal and distal operator sites are too strong). We have relegated the remainder of the model to the SI, and we believe that the main text section is both clearer and more focused on the overall goal of the project, to identify characteristics of an optimal promoter.

Reviewer #2 (Remarks to the Author):

This article presents the employment of a rationally designed approach to study the effect of both RNA polymerase binding and LacI elements in various compositions and architectures within

IPTG induced promoters in bacteria. The authors utilized an MPRA workflow in order to allow the parallel examination of over 8,200 combinatorial options. The authors performed a thorough characterization of both ends of promoter activity, i.e repression and transcription, and utilize their high-throughput data to build a model predicting in silico the best possible outcome – the strongest possible minimal leaky IPTG inducible promoter. Later they construct libraries with various changes to the basic architecture in order to tackle their role in driving minimal baseline expression and the highest possible fold-change (induction). Overall, the paper is well written, easy to follow the scientific logic and could be a good fit for Nature Communications after revision. I have the following comments:

Major points

-In the introduction and discussion, would be good to mention and discuss what has already been done in this space for eukaryotic regulatory elements and how it compares to this work. There are several relevant articles on this that the authors do not mention. Some of these that come to mind are: PMIDs 31792407, 23921661, 2389260, 22609971, 31253799.

We thank Reviewer 2 for making this point. We have updated the introduction to include similar work previously performed using eukaryotic regulatory elements and how it relates to this work.

- Inspired by previous success in studying the combinatorial logic of *E. coli* promoters²⁰, we sought to address these obstacles by integrating rational design with high-throughput screening of large DNA-encoded libraries. The recent development of massively-parallel reporter assays (MPRAs) provides a framework for leveraging next-generation sequencing to measure cellular transcription levels of large numbers of DNA sequence variants. ***Previously, this paradigm has also been used to empirically examine both the individual and combinatorial effects of transcription factor binding sites on gene expression in eukaryotes, improving our ability to design synthetic eukaryotic promoters with programmable responses²¹⁻²⁸. However, there have been few similar high-throughput studies in prokaryotes.*** Here, we implemented a genomically-encoded MPRA to interrogate thousands of rationally designed variants of the *lacZYA* promoter and investigate relationships between inducible promoter components across four *cis*-regulatory sequence architectures.

Additionally we have added a section to the discussion referencing previous work in eukaryotes and how our work fits into this larger narrative:

- Beyond studying combinatorial effects within the *P_{lacZYA}* architecture, we investigated these interactions in alternative promoter contexts. Characterizing the dynamic range of expression of novel inducible promoter architectures expands our ability to fine-tune metabolic pathways for generating chemical compounds, especially when products are toxic to the host system⁶⁸. Furthermore, this approach could be applied towards

identifying design rules to minimize leakiness and maximize fold-change in other bacterial repressor systems which likely operate under similar thermodynamic principles. To our knowledge, a systematic analysis exploring a similar interplay between TF and core promoter strengths in eukaryotic systems has yet to be performed, however many MPRA have explored the regulatory role of TFs^{22-28,69} and core promoter⁷⁰ binding site compositions separately. Thus we predict the approach presented here can inform us about the interactions between TF and core promoter sites in other systems.

-It would be good if the authors discuss how they think their results implicate other E. coli or bacterial promoters. Are they universal or do they think there will be different rules governing function there. How do they compare to eukaryotic ones, i.e. the above examples. Also, how they could be further used in research or industry for example, for the generation of compounds etc.

We have added an additional section in the discussion addressing the universality of our findings to bacterial and eukaryotic promoters. We have also contributed a discussion on how our approach could benefit research or industrial applications:

- Characterizing the dynamic range of expression of novel inducible promoter architectures expands our ability to fine-tune metabolic pathways for generating chemical compounds, especially when products are toxic to the host system⁶⁸. Furthermore, this approach could be applied towards identifying design rules to minimize leakiness and maximize fold-change in other bacterial repressor systems which likely operate under similar thermodynamic principles.

Minor points

-For the first results paragraph, 'The lacZYA promoter...', Would be good to have a main figure showing everything mentioned there so easy for the reader to understand the structure/function of this promoter.

Thank you for the suggestion, which we agree improves the accessibility of the work. We have updated figure 1 to include a schematic of the canonical *lacZYA* promoter:

Figure 1) Identifying optimal spacing for repressors at *lacUV5* promoter. **A)** We designed a library of *lacUV5* variants modeled after the WT *lacZYA* promoter. In this library, we evaluate repressor effects when the *distal* site is moved 32 nucleotides upstream at 1 bp increments. **B)** If repressors bind along the same face of the DNA helix, repression loop formation may occur, thereby preventing RNAP association with the promoter. **C)** In this MPRA format, pooled promoter variants are engineered to express uniquely barcoded sfGFP transcripts, singly integrated into the *essQ-cspB* locus of the *E. coli* genome. After integration, individual promoter expression was determined *en masse* using the ratio of barcode reads from RNA-seq to that of DNA-seq. **D)** Comparison of MPRA expression measurements between biological replicates grown in MOPS rich-defined medium supplemented with 0.2% glucose ($r = 0.987$, $p < 2.2 \times 10^{-16}$). **E)** MPRA expression when a *proximal* site is added relative to expression of *lacUV5* without repressor sites. Top shows the distribution of expression for all barcodes associated with each variant ($n \geq 35$ for all TFs) whereas bottom shows the averaged variant expression relative to *lacUV5* without repressor site (null). **F)**

Relative MPRA expression as each *distal* site is moved upstream in the absence of a *proximal* site relative to *lacUV5* without repressors. Thick lines denote the fit using locally weighted polynomial regression. Thin lines connect data points at sequential intervals. Gray bars indicate 3 bp windows where the *distal* site is positioned in-phase with the +11 *proximal* site¹⁷. **G**) MPRA expression as the *distal* site is moved upstream when the *proximal* site is present relative to expression of the *proximal*-only variant. Source data are provided as a Source Data file.

Additionally, we have added a reference to the *PlacZYA* architecture and looping mechanism in figure 1A earlier in the text.

- The *lacZYA* promoter is a classic model for gene regulation in *E. coli*, with many studies investigating the relationship between sequence composition and induction properties. This promoter contains two LacI dimer sites positioned at the *proximal* +11 and *distal* -82 positions relative to the transcription start site (TSS)^{30,31}, which flank a set of $\sigma 70$ -10 and -35 elements (**Figure 1A, see WT *PlacZYA***).

-In the first three results segments, most of the P values are 2.2×10^{-16} , assuming likely that this is the lowest you can go, but need to explain why.

To clarify the use of this value, we have added a sentence to the ‘Code Availability, Data Availability, & Data Analysis’ section of the text:

Statistical significance is reported to a lower limit of $p < 2.2 \times 10^{-16}$, the lowest reportable value by R.

-Line 124 – The authors spent two paragraphs establishing the mechanism of operation of the distal and proximal LacI binding sites, and then just jumped to LacUV5 promoter. Might be due to a gap in my knowledge of E. Coli operons, but I believe a connection should be made between segments

To clarify the connection between the *lacZYA* promoter and the *lacUV5* promoter, we have modified the following section:

Accordingly, we tested the relationship between spacing and repression for six transcription factors (TFs) at **the most commonly utilized *lacZYA*-derived promoter**, *lacUV5*: LacI, AraC, GalR, GlpR, LldR, and PurR. While LacI^{35,37,39}, AraC^{40,41} and GalR^{42–46} have been experimentally shown to engage in DNA looping, there is evidence that GlpR⁴⁷, LldR⁴⁸, and PurR³⁴ may also be capable of this mechanism. Using reported, natural binding sites for these TFs⁴⁹ (**Table S1**), we designed 624 sequences assessing the ability of these sites to repress a constitutive *lacUV5* promoter across various operator spacings. **The *lacUV5* promoter models the *lacZYA* canonical**

architecture, but has a small 2bp mutation in the -10 to drive more detectable levels of expression⁵⁰.

-Line 146-147 – the number format on the p value is wrong, replace $p < 2.2 \times 10^{16}$ to $p < 2.2 \times 10^{-16}$, if the corrected version is what the authors meant.

This section has been corrected to say:

- These measurements exhibited a high degree of correlation between technical replicates (**Figure 1D**, $r = 0.987$, $p < 2.2 \times 10^{-16}$).

-Line 148-153 – This entire segment that relates to Figure 1d seems like an experiment that was not performed as an MPRA. If so, please correct the text to represent this. Additionally, add significance stats to the text.

We realize this section may have implied the experimental results were obtained outside of the MPRA. The legend for this figure (now 1E), which this section discusses, has been revised to say:

- **MPRA** expression when **a proximal** site is added relative to expression of *lacUV5* without repressor sites.

In addition, we have performed an additional analysis to show the significant difference in expression between these promoter variants. We measure the expression of each promoter by evaluating the expression of multiple barcodes (median n=70 barcodes per variant) and perform a t.test between barcode measurements (Figure 1E, above).

-Lines 159-160 – Authors claims “uniformity of response”, it might be beneficial to be more specific and state “uniformity of cyclic behavior”.

This section has been revised to say:

- We observed a uniformity of **cyclic behavior** across most repressors tested, suggesting position-dependent effects are a general phenomenon of many TFs.

-Lines 159-165 – If not because of DNA looping, please mention briefly an explanation to the alternating behavior while lacking the proximal site, meaning just based on the distance from the TSS.

We thank both reviewers for bringing the need to further discuss this interesting finding to our attention. Please see our response to Reviewer 1:

We thank Reviewer 1 for highlighting this section and motivating us to further investigate these interesting observations. We now investigate the nature of these oscillations and how they relate to the helical phasing between *proximal* and *distal* operator sites. Specifically, we identified the distance between operators where the center of the *distal* site is in phase with the +11 *proximal* sites and by extension the +1 transcription start site (see grey bars below). This shows us interesting differences between the behaviors and mechanism of action of these transcription factor binding sites.

Notably, in the absence of a proximal site, GalR and PurR *distal* sites repress transcription when in phase (grey bars) with the +1 position yet increase expression when out of phase (Figure 1F). On the other hand, GlpR *distal* sites result in activation when in phase with the

We have expanded the analysis on TF expression oscillation in the main text, reproduced below:

- First, we looked at the effect of moving the *distal* site across 33 nucleotides in the absence of a functional *proximal* site (Figure 1F). We observed a uniformity of cyclic behavior across most repressors tested, suggesting position-dependent effects are a general phenomenon of many TFs. Lone GalR, GlpR, PurR *distal* sites alternated between activation and repression, a phenomenon which has been observed in similar translocations of a LacI binding site upstream of a promoter in the absence of inducer³⁷. **This may be due to helical positioning of the repressor relative to RNA polymerase and resulting steric interference, or binding-induced DNA distortions⁵³. Interestingly, we observed TFs exhibiting opposing position-dependent behaviors, where GalR and PurR repressed when the center of the binding sites were approximately in-phase with the +1 TSS position and activated when out of phase. We observed the opposite effect with GlpR sites.**

In addition, we include this statement to acknowledge the differences in loop-dependent activities and propose hypotheses for further investigation:

- Additionally, *distal* site and loop-mediated repression differed between TFs tested which is likely due to differences in how these proteins are situated on their binding sites or oligomerize to form DNA loops.

While we think these findings warrant further investigation, especially to learn the mechanisms by which *distal* transcription factors can influence expression, we feel this would be beyond the scope of the paper, which intends to learn the relationship between DNA sequence structures and promoter expression.

-Figure 1c – Axis names, add units –

The figure axes have been altered to indicate measurements are in units of (RNA/DNA). See revised figure 1 (Now figure 1D) above.

-Figure 1d – Clear from the text and the legend but for a better visual representation worth adding another bar in black to represent the NULL proximal reference point. Also, lacking error-bars, significance scores and n (if different from the already stated n=2 for the MPRA, Figure 1c).

We thank the reviewer again for the suggestions to improve the clarity and confidence in the analysis. We have 1) Included the data and bar for the null *proximal* variant (i.e. *lacUV5* without repressor). 2) We have added boxplots comparing the distribution of barcode measurements per variant (See response above for explanation). 3) Updated the legend to state the number of observations in this figure:

E) MPRA expression when a *proximal* site is added relative to expression of *lacUV5* without repressor sites. Top shows the distribution of expression for all barcodes associated with each variant ($n \geq 35$ for all TFs) whereas bottom shows the averaged variant expression relative to *lacUV5* without repressor site (null).

One more point, the illustration exhibit the increment movement of the distal site, which does not seem to be part of the this experiment; correct illustration in accordance.

All illustrations have been updated to clarify the variants being compared in each analysis. See revised figure above.

-Figure 1e-f – representation of the results with an increasing distance going left-to-right, while the illustration shows the distal element going right-to-left might be confusing. Consider changing

the x-axis to either flipped order to indicate the location of the distal element to match the illustration.

The X-axes in these figures have been flipped to match illustrations. See revised figure above.

-Figure 1e-f – if possible to add visual representation of when the proximal and distal sites are in phase or out of phase (a shade or faint borders).

We have added grey bars to figures 1F and 1G to represent when the *distal* sites are in phase with the *proximal* sites. We use the results of [citation] et al., which showed that LacI sites are in phase at 115.5 bp and with ~11.5 bp oscillations. Based on this prior work, we considered transcription factors in-phase if they are within a 3bp window centered at 104 bp, 115.5 bp, and 127 bp apart.

-Line 216 – Table S2 is mentioned prior to Table S1 (in material and methods), switch if needed based on final paper outline.

We have updated the text to refer to Table S1 earlier in the text:

- Using reported, natural binding sites for these transcription factors⁴¹ (**Table S1**), we designed 624 sequences assessing the ability of these sites to repress a constitutive *lacUV5* promoter across various operator spacings.

-Lines 217-218 – unnecessary line/paragraph break

Line break removed.

-Line 259-260 – be consistent of formatting the fold change numbers either #-fold, just the # or #x.

We have formatted all fold-change numbers to “#-fold” for consistency:

- Notably, while the consensus core promoter containing O_{sym} in both the *proximal* and *distal* sites yielded a change of **4.63**, its counterpart containing the weaker O_1 variant in the *proximal* site drove an increased fold-change of **8.97**.

-Line 324 – be consistent with the format of R2. Either $R2=0.79$ or $R2=.79$

We have corrected the text to say:

- Enforcing this previous value for $p_{act}^{repressor}$ while fitting the model resulted in comparable parameter values (Figure S4C) and overall fit ($R^2 = 0.79$, $p < 2.2 \times 10^{-16}$).

-Line 353-355 – why the authors choose to include another upstream site of the repressor? Was that indicated from their model? If so they should explain this gap in knowledge.

We have updated the text to elaborate on our rationale:

- In particular, we expected that an additional distal site would enhance repression, **as multiple upstream sites would increase the probability of repressor binding and loop formation.**

-Line 355 and Figure 4A – from the text I would expect to see a larger result matrix to include 5x distal and 5x proximal and the combination with 4x -10 and -35 sites. The matrix presents only the proximal-distal 25 combinations. Either reflect correctly in figure or revise the text.

We have clarified in the text that this figure (4A, Bottom) is examining the effect of the *distal+* site in combination with each *distal* variant within the context of promoters with consensus -10 and -35 elements and O_1 or O_{sym} in the proximal *lacI* site:

- We limited our analysis to studying promoters with consensus core promoter elements as well as an O_1 or O_{sym} *proximal* site to best capture the repressive effects of the *distal+* element.

-Figure 4C + lines 433-435 – Authors indicate that they changed the location of the proximal site from +11 to +30. This is not indicated in the illustration.

We have updated the PSteric library schematic in Figure 4C to reflect this, where proximal sites were either positioned at canonical +11 or shifted to +30. See figure.

Figure 4) Optimizing alternative IPTG-inducible promoter architectures. **A) Top:** Design for Pmultiple library. **Bottom:** The average effect of the *distal+* site (rows) on fold-change given the *distal* site identity (column). Here we examine consensus -10/-35 promoters containing O_1 or O_{sym} in the *proximal* site. **B) Top:** Design for Pspacer library. **Bottom:** Comparison of uninduced expression, induced expression, and fold-change between variants composed of the same sequence elements in the Pspacer and Pcombo architectures (Two-sided Mann-Whitney U-tests). We examined only active promoters containing a consensus -10 and/or -35 sequence. **C) Top:** Psteric library design. **Bottom:** The fold-change of promoters containing O_1 in both the *core* and *proximal* sites and a 56 bp inter-operator distance. Here we examine the effect of the -10 element in conjunction with the strongest UP and extended -10 element combinations. N/A indicates data missing from our analysis. **D)** Distributions of uninduced expression, induced expression, and fold-change for variants with fold-change ≥ 2 in each library. Dashed line separates active from inactive sequences and is set as the median of the negative controls + $2 \times$ median absolute deviation (Two-sided Mann-Whitney U-tests with Benjamini-Hochberg correction, '*' = $p < 0.05$, '**' = $p < 0.01$, '***' = $p < 0.001$). Source data are provided as a Source Data file.

-Figure 4C – the UP element(-) AND Extended -10 (+) group is missing a bar (TATAAA)

Figure 4C has been updated with N/A at TATAAA UP (-), Extended -10 (+) to clarify that this data was missing from our analysis. This clarification has also been added to the legend text. See figure above.

-Lines 448-449 – Authors mention highest variants (from Figure S7B), I would add a note that overall the median FC for all variants was not significantly shifted. Additionally, I think a violin plot will work better for Figure S7B.

We have updated the text to clarify that:

- First, we found variants with the highest fold-change were constructed with *proximal* operator sites located at the +30 position relative to the TSS, though the overall median fold-change of promoters did not differ between the two proximal operator site positions (Figure S7B).

Additionally, we have changed Figure S7B to a violin plot. See revised figure.

-Lines 450-452 – It is not clear rather the authors are related to the +30 Proximal site highest FC promoters, or all promoters (+11 and +30). Please clarify in text

We have updated the text to clarify that we are exclusively considering promoters with a +30 Proximal site:

- When all three are present, promoters containing a *proximal* operator site located at the +30 position exhibit up to a 11.8-fold response to IPTG (Figure 4C, bottom).

-Lines 466-468 – The authors did not relate to the Pcombo vs. Pspacer similar uninduced expression. Overall from Figure 4D it seems that either pcombo or Pspacer behave similarly. Adding a note on that in the text will be good.

Thank you for pointing out this inconsistency. It appears we lose this relationship because Figure 4D only compares variants from each library with fold-change > 2. We have updated the text to say:

- Although previously we found Pspacer variants exhibited greater uninduced and induced expression than Pcombo variants, we did not observe this phenomenon between these subsets of each library.

-Line 476 – change “four libraries” to “four architectures”

We have updated the text to say:

- From all **four architectures**, we individually evaluated promoter sequences exhibiting higher fold-change with low leakiness by using flow-cytometry to measure sfGFP expression in uninduced (0mM IPTG) and fully induced (1mM IPTG) conditions.

-Figure S9 – It would be better if all y-axis be similar.

The figure axes have been altered such that the y-axis range is the same for each panel, in units of $\log_2(\text{Fold-change})$. See revised figure.

Additional changes:

Line 492: **We have shortened this supplementary section where we use fluorescent reporters to validate our MPRA measurements to be more concise**

Reviewers' Comments:

Reviewer #1:

Remarks to the Author:

I thank the authors for taking the time and answering my previous comments. However, this work still falls short, in my opinion, of the standard needed for publication in Nature Communications. There have been many MPRA papers published in recent years that provide detailed description of the parts being studied. However, nowadays, the real test of MPRA is not only in generating large data sets but also in providing a new mechanistic understanding. This can be achieved in one of two ways:

1. Provide evidence for a new regulatory phenomenon, and then follow it up with some evidence for a mechanism. In this paper, the authors MPRA did reveal a new regulatory phenomenon (as mentioned in my previous review), but there were no follow-up experiments that attempted to tease out the underlying mechanism.

2. Alternatively, if one uses some underlying theoretical scheme to model the data, validation of the model should be carried out on predicted or "unseen" sequences (see for example de Boer et al., Nat Biotechnology 2019). In the present revision, the verifications were carried out only on variants that were already present in the OL (Figure 5 – if I understood correctly). In addition, since the authors chose a thermodynamic modeling approach, it would be interesting if they attempted to design, model, and then measure variants with untested promoters (say from their 2018 paper), different combinations of binding sites (i.e. more than one TF), and different numbers of sites (i.e. more than 2). While a verification effort such as was carried out by the de Boer paper is not necessarily needed, a set of composite promoters as suggested above that directly test the model's prediction will satisfy this criterion, even (and especially if) success rate will not be 100%.

Reviewer #2:

Remarks to the Author:

The authors have nicely addressed all my comments.

We are grateful for the opportunity to revise our work and improve upon the scope of our findings. In particular, we have implemented the reviewer's suggestion to validate our thermodynamic model on unseen sequences as well as on promoters using a different architecture. Specifically, we now show that 1) When trained on as little as 5% of the data for promoters consisting of a single architecture, the model is as capable of predicting the induced and uninduced expression of the remaining 95% of unseen promoters with the same architecture as well as a model trained on the full suite of data and 2) The model can be adapted to predict the expression of promoters with a different architecture, especially when each modular piece has been previously characterized. We believe that these findings demonstrate the power and robustness of our model approach and provide insights into potential strategies for creating generalizable models for predicting bacterial promoter transcriptional dynamics. We have left the reviewer's remarks in italics and have included our response in regular font. We have used indented sections to indicate text that has been added to the manuscript.

=====

Reviewer #1 (Remarks to the Author):

I thank the authors for taking the time and answering my previous comments. However, this work still falls short, in my opinion, of the standard needed for publication in Nature Communications. There have been many MPRA papers published in recent years that provide detailed descriptions of the parts being studied. However, nowadays, the real test of MPRA is not only in generating large data sets but also in providing a new mechanistic understanding. This can be achieved in one of two ways:

1. Provide evidence for a new regulatory phenomenon, and then follow it up with some evidence for a mechanism. In this paper, the authors MPRA did reveal a new regulatory phenomenon (as mentioned in my previous review), but there were no follow-up experiments that attempted to tease out the underlying mechanism.

2. Alternatively, if one uses some underlying theoretical scheme to model the data, validation of the model should be carried out on predicted or "unseen" sequences (see for example de Boer et al., Nat Biotechnology 2019). In the present revision, the verifications were carried out only on variants that were already present in the OL (Figure 5 – if I understood correctly). In addition, since the authors chose a thermodynamic modeling approach, it would be interesting if they attempted to design, model, and then measure variants with untested promoters (say from their 2018 paper), different combinations of binding sites (i.e. more than one TF), and different numbers of sites (i.e. more than 2). While a verification effort such as was carried out by the de Boer paper is not necessarily needed, a set of composite promoters as suggested above that directly test the model's prediction will satisfy this criterion, even (and especially if) success rate will not be 100%.

We appreciate the reviewer's interest in pushing the boundaries of this work. We were particularly intrigued by the suggestion of using our thermodynamic model to predict the expression of unseen sequences, rather than simply explanation of the trends in our current dataframe. To that end, we

performed two additional analyses to evaluate the model's ability to predict expression by 1) Separating the training and testing sequences and 2) Translating these results into a different architecture. For the first point, we find that using very few sequences (as little as 5% of our promoters) to train the model enables us to predict the expression of the remaining promoters as accurately as when training the model on 90% of the data. For the second point, we extended the modeling results from our simplest four-component architecture with (Lacl_{Distal}-RNAP₋₃₅-RNAP₋₁₀-Lacl_{Proximal}) to the five-components architecture consisting of (Lacl_{Distal+}-Lacl_{Distal}-RNAP₋₃₅-RNAP₋₁₀-Lacl_{Proximal}). We felt that this was the most important extension to test, since we could utilize the previously-determined binding energies of each component (note that the Lacl_{Distal+} and Lacl_{Distal} sequences in the new architecture are identical and hence have the same binding energies) to predict the *entire* suite of 1600 gene expression measurements without resorting to any fitting.

1) Firstly, instead of fitting our model parameters using our entire data set, we trained the model on a subset of data and predicted the expression of the remaining sequences within the same architecture. We previously demonstrated that the thermodynamic model parameter values trained using the whole library of Pcombo variants could accurately fit the data ($R^2 = 0.79$, $p < 2.2e^{-16}$). In this analysis, we sampled various subsets of the data multiple times, from 1% to 90% of the variants, used these subsets to fit all model parameters, and predicted expression of the remaining promoters (see figure). We were surprised to find that relatively few promoters were needed to train all model parameters, and that as little as 5% of the library was sufficient to fit binding energies as accurately as when trained on the whole library. Therefore, we demonstrate that this model is able to accurately predict the gene expression of unseen sequences (following the same architecture) even when trained on relatively small amounts of data.

We have made the following changes to the main text to describe these new findings:

Using this statistical mechanics model of gene expression, we inferred the binding energies of each promoter element and compared the resulting fits for the 1,493 different promoters in the absence of IPTG (**Figure 3A**, $R^2 = 0.79$, $p < 2.2 \times 10^{-16}$, parameter values in **Figure S4B**). **Interestingly, we found that all parameters could be fit using as little as 5% of the library and retain the ability to accurately predict the other 95% of variants when used in this model framework (Figure S5A)**. Furthermore, this model enables us to extrapolate the gene expression for promoter architectures with arbitrary binding strengths spanning the theoretical parameter space (**Figure 3B**).

2) Furthermore, we have taken the reviewer's suggestion to verify the model on a different sequence architecture. We evaluated whether our model, which accurately predicts variant expression with the Pcombo architecture, could also predict variant expression from the Pmultiple library which includes an additional *distal* Lacl binding site (called *distal+*). We extended the statistical mechanics model to include the additional states available to this architecture due to the presence of an additional *distal+* binding site and predicted expression of each variant using parameter values fit using the Pcombo variants. We were surprised to see that the model performed relatively well when challenged with these previously unseen sequences of a different architecture (see Figure, $R^2 = 0.62$, $p < 2.2e^{-16}$). While this correlation is lower than the $R^2=0.79$ observed with the Pcombo library, we stress that each point in the figure represents a pure prediction using previously established thermodynamic parameters. This demonstrates the adaptability of our model framework, wherein all one needs to do is determine the available states for any given sequence architecture and plug in binding energy parameter values to accurately predict expression of *lacUV5* variants.

We have added an additional section to the text describing the implementation of this analysis.

*Finally, we explored whether our previously established statistical mechanics model could accurately predict expression of variants in this library. We extended our model framework to account for the different promoter states available to the Pmultiple architecture (Described in Supplementary Methods) while retaining the same parameter values fit to the Pcombo library. Despite a lack of training on promoters of this architecture, the model was still able to predict expression of Pmultiple variants with impressive accuracy (Figure S5B, $R^2 = 0.62$, $p < 2.2 \times 10^{-16}$). We expect the drop in accuracy is related to the observed interactions between the *distal* and *distal+* sites, which will require further studies to parameterize. Nonetheless, we show that this adaptable model framework is robust even across previously unseen sequence architectures.*

Also, we have a more in-depth description of how we extended the thermodynamic model in the supplementary appendix.

Extending the model framework for the Pmultiple architecture

To extend our equation for gene expression to predict expression of Pmultiple promoters, we modified our equation for gene expression to consider the additional states that would be possible given the presence of a *distal+* Lacl site. For simplicity, we assume that Lacl cannot be simultaneously bound to both the *distal* and *distal+* site given that both sites are immediately adjacent to one another, and hence relatively few additional states need to

be introduced into the model. Below is a comparison of the original and modified equations:

Without Distal+ $\text{Gene Expression} = \frac{r_{\min}(Z_{\text{notProx}} + Z_{\text{prox}} + e^{-\beta(E_{-35} + E_{-10})} Z_{\text{prox}}) + r_{\max} e^{-\beta(E_{-35} + E_{-10})} Z_{\text{notProx}}}{(1 + e^{-\beta(E_{-35} + E_{-10})})(Z_{\text{prox}} + Z_{\text{notProx}})}$ where Z_{prox} represents the sum of weights for all states where the proximal site is bound while Z_{notProx} equals the sum of weights for all states where the proximal site is not bound, namely, $Z_{\text{prox}} = e^{-\beta E_{\text{prox}}} + e^{-\beta(E_{\text{prox}} + E_{\text{dist}})} + e^{-\beta(E_{\text{prox}} + E_{\text{dist}} + E_{\text{loop}})}$ $Z_{\text{notProx}} = 1 + e^{-\beta E_{\text{dist}}}$	With Distal+ $\text{Gene Expression} = \frac{r_{\min}(Z_{\text{notProx}} + Z_{\text{prox}} + e^{-\beta(E_{-35} + E_{-10})} Z_{\text{prox}}) + r_{\max} e^{-\beta(E_{-35} + E_{-10})} Z_{\text{notProx}}}{(1 + e^{-\beta(E_{-35} + E_{-10})})(Z_{\text{prox}} + Z_{\text{notProx}})}$ where Z_{prox} represents the sum of weights for all states where the proximal site is bound while Z_{notProx} equals the sum of weights for all states where the proximal site is not bound, namely, $Z_{\text{prox}} = e^{-\beta E_{\text{prox}}} + e^{-\beta(E_{\text{prox}} + E_{\text{dist}})} + e^{-\beta(E_{\text{prox}} + E_{\text{dist}}^+)} + e^{-\beta(E_{\text{prox}} + E_{\text{dist}} + E_{\text{loop}})} + e^{-\beta(E_{\text{prox}} + E_{\text{dist}}^+ + E_{\text{loop}})}$ $Z_{\text{notProx}} = 1 + e^{-\beta E_{\text{dist}}} + e^{-\beta E_{\text{dist}}^+}$
---	---

Notably, the equation for gene expression remains unchanged and the only difference is what states are within Z_{prox} (which sums over all states where the promoter is repressed), and Z_{notProx} (where the proximal site is unbound and the promoter will still be active). Specifically, the terms in the modified Z_{prox} represent the states where only the *proximal* Lacl site is bound, the *proximal* and *distal* sites are bound, the *proximal* and *distal*+ sites are bound, the *proximal* and *distal* sites are bound and loop the DNA, and that the *proximal* and *distal*+ sites are bound and loop the DNA. On the other hand, Z_{notProx} has three terms representing that neither the *distal*+ nor *distal* sites are bound (1), that only the *distal* site is bound ($e^{-\beta E_{\text{dist}}}$), and that only the *distal*+ site is bound ($e^{-\beta E_{\text{dist}}^+}$). Relaxing the assumption that Lacl could not be bound to both *distal* and *distal*+ sites did not affect the resulting R^2 .

Both figures presented in this response have been included as supplementary figures:

Figure S5) Predictive modeling of unseen lacUV5 promoter variants. A) Correlation between predicted and actual expression levels when different proportions of data are used as training sets to approximate model parameters. For each proportion, twenty unique, randomized samplings of the Pcombo library were used as training input and the remaining Pcombo promoters were used for prediction. **B)** Thermodynamic parameter values fit to the Pcombo library expression (**Figure S4B**) enable moderate ability to predict induced and uninduced expression of Pmultiple variants when used in an adapted thermodynamic model.

Although the reviewer requested we design a new library of variants, model their expression, and validate these predictions, this is unfortunately unfeasible for us. Instead we saw the opportunity to perform an equivalent assessment of our model using the various libraries we have already characterized here. In our opinion, the value of a model is tied to 1) whether it can explain an appreciable % of the variance in expression of a set of sequences and 2) whether the framework is interpretable and posits generalizable biological mechanisms. We now demonstrate that the thermodynamic model presented in this work can predict the expression of previously unseen sequences with relatively high fidelity across two promoter architectures and therefore meets both these criteria. Furthermore, due to the accuracy and mechanistic framework of our model, we may use it to make verifiable predictions of the behavior of these inducible systems. For instance, the model indicates a range of LacI binding energies to maximize fold-change in expression of Pcombo variants and also suggests the phenomenon that promoters with weaker RNAP binding sites require weaker LacI sites to achieve optimal fold-change. Both of these predictions are supported by observations in our data. Therefore, we conclude that this model represents a useful theoretical scheme for LacI-regulated promoters which may be applicable to other repressor systems in bacteria.

Reviewer #1 (Remarks to the Author):

Thank you for expanding the model the other OL, and showing its predictive ability on an alternative system. As the authors mentioned, I asked for new experiments that will break away from the limited LacI system that they studied to really examine the broader applicability of their approach. While I did not request another OL, and would have been satisfied with a small set of low-throughput results, I do understand the limitations the the current COVID-19 predicament created for the authors. But there is a price. A more extensive validation strategy could have made their algorithm useful to the broader Synthetic Biology community. In addition, it would have provided a more valuable insight into TF interaction, looping, and over-all bacterial promoter activity as compared with the current limited scope.

Given the above, I leave the decision to the editor whether to accept this paper or not.

Reviewer #1 (Remarks to the Author):

Thank you for expanding the model to the other OL, and showing its predictive ability on an alternative system. As the authors mentioned, I asked for new experiments that will break away from the limited LacI system that they studied to really examine the broader applicability of their approach. While I did not request another OL, and would have been satisfied with a small set of low-throughput results, I do understand the limitations the current COVID-19 predicament created for the authors. But there is a price. A more extensive validation strategy could have made their algorithm useful to the broader Synthetic Biology community. In addition, it would have provided a more valuable insight into TF interaction, looping, and over-all bacterial promoter activity as compared with the current limited scope.

Given the above, I leave the decision to the editor whether to accept this paper or not.

Author Response:

We are very appreciative of your constructive feedback which has enriched the paper and looks toward the future of this area of research. Beyond the logistical difficulties of performing further experiments at this time, a set of well-designed low-throughput results would not be feasible as it would require extensive characterization or advanced knowledge of the binding kinetics of another OL. This is because our model framework relies on *a priori* knowledge or imputation of the binding energies of operators in our experimental system. We do believe that there is a great wealth of knowledge to be attained using this approach to break into other OL systems, however, this would require a dedicated effort beyond the scope of this work.